



# Effect of ocean acidification and elevated $f$CO$_2$ on trace gas production by a Baltic Sea summer phytoplankton community

**A.L. Webb[1,2], E. Leedham-Elvidge[1], C. Hughes[3], F.E. Hopkins[4], G. Malin[1], L.T. Bach[5], K. Schulz[6], K. Crawfurd[7], C.P.D. Brussaard[7,8], A. Stuhr[5], U. Riebesell[5], and P.S. Liss[1].**

[1] {Centre for Ocean and Atmospheric Sciences, School of Environmental Science, University of East Anglia, Norwich, UK, NR4 7TJ}

[2] {Groningen Institute for Evolutionary Life Sciences, University of Groningen, 970 CC Groningen, The Netherlands}

[3] {Environmental Department, University of York, York, UK, YO10 5DD}

[4] {Plymouth Marine Laboratory, Plymouth, UK, PL1 3DH}

[5] {GEOMAR Helmholtz Centre for Ocean Research Kiel, Düsternbrooker Weg 20, 24148 Kiel, Germany.}

[6] {Centre for Coastal Biogeochemistry, School of Environment, Science and Engineering, Southern Cross University, Lismore, NSW 2480, Australia.}

[7] {Department of Biological Oceanography, NIOZ – Royal Netherlands Institute for Sea Research, PO Box 59, 1790 AB Den Burg, Texel, The Netherlands}

[8] {Aquatic Microbiology, Institute for Biodiversity and Ecosystem Dynamics, University of Amsterdam, P.O. Box 94248, 1090 GE, Amsterdam, The Netherlands}

Correspondence to: Alison Webb (a.l.webb@rug.nl)

**Abstract**

**The Baltic Sea is a unique environment as the largest body of brackish water in the world. Acidification of the surface oceans due to absorption of anthropogenic CO$_2$ emissions is an additional stressor facing the pelagic community of the already challenging Baltic Sea. To investigate its impact on trace gas biogeochemistry, a large-scale mesocosm experiment was performed off Tvärminne Research Station, Finland in summer 2012. During the second half of the experiment, dimethylsulphide (DMS) concentrations in the highest $f$CO$_2$ mesocosms (1075 -**



**1333 µatm) were 34% lower than at ambient $CO_2$ (350 µatm). However the net production (as**
**measured by concentration change) of seven halocarbons analysed was not significantly affected**
**by even the highest $CO_2$ levels after 5 weeks exposure. Methyl iodide ($CH_3I$) and diiodomethane**
**($CH_2I_2$) showed 15% and 57% increases in mean mesocosm concentration ($3.8 \pm 0.6$ pmol $L^{-1}$**
**increasing to $4.3 \pm 0.4$ pmol $L^{-1}$ and $87.4 \pm 14.9$ pmol $L^{-1}$ increasing to $134.4 \pm 24.1$ pmol $L^{-1}$**
**respectively) during Phase II of the experiment, which were unrelated to $CO_2$ and corresponded**
**to 30% lower Chl-*a* concentrations compared to Phase I. No other iodocarbons increased or**
**showed a peak, with mean chloroiodomethane ($CH_2ClI$) concentrations measured at $5.3$ ($\pm 0.9$)**
**pmol $L^{-1}$ and iodoethane ($C_2H_5I$) at $0.5$ ($\pm 0.1$) pmol $L^{-1}$. Of the concentrations of bromoform**
**($CHBr_3$; mean $88.1 \pm 13.2$ pmol $L^{-1}$), dibromomethane ($CH_2Br_2$; mean $5.3 \pm 0.8$ pmol $L^{-1}$) and**
**dibromochloromethane ($CHBr_2Cl$, mean $3.0 \pm 0.5$ pmol $L^{-1}$), only $CH_2Br_2$ showed a decrease of**
**17% between Phases I and II, with $CHBr_3$ and $CHBr_2Cl$ showing similar mean concentrations in**
**both Phases. Outside the mesocosms, an upwelling event was responsible for bringing colder,**
**high $CO_2$, low pH water to the surface starting on day $t16$ of the experiment; this variable $CO_2$**
**system with frequent upwelling events implies the community of the Baltic Sea is acclimated to**
**regular significant declines in pH caused by up to 800 µatm $fCO_2$. After this upwelling, DMS**
**concentrations declined, but halocarbon concentrations remained similar or increased compared**
**to measurements prior to the change in conditions. Based on our findings, with future**
**acidification of Baltic Sea waters, biogenic halocarbon emissions are likely to remain at similar**
**values to today, however emissions of biogenic sulphur could significantly decrease from this**
**region.**

## 1    Introduction

Anthropogenic activity has increased the fugacity of atmospheric carbon dioxide ($fCO_2$) from 280
µatm (pre-Industrial Revolution) to over 400 µatm today (Hartmann *et al.*, 2013). The IPCC AR5
long-term projections for atmospheric $pCO_2$ and associated changes to the climate have been
established for a variety of scenarios of anthropogenic activity until the year 2300. As the largest
global sink for atmospheric $CO_2$, the global oceans have absorbed an estimated 30% of excess $CO_2$
produced (Canadell *et al.*, 2007). With atmospheric $pCO_2$ projected to possibly exceed 2000 µatm by
the year 2300 (Collins *et al.*, 2013; Cubasch *et al.*, 2013), the ocean will take up increasing amounts of
$CO_2$, with a potential lowering of surface ocean pH by over 0.8 units (Raven *et al.*, 2005). The overall
effect of acidification on the biogeochemistry of surface ocean ecosystems is unknown and currently




unquantifiable, with a wide range of potential positive and negative impacts (Doney *et al.*, 2009;
Hofmann *et al.*, 2010; Ross *et al.*, 2011).
A number of volatile organic compounds are produced by marine phytoplankton (Liss *et al.*, 2014),
including the climatically important trace gas dimethylsulphide (DMS, $C_2H_6S$) and a number of
halogen-containing organic compounds (halocarbons) including methyl iodide ($CH_3I$) and bromoform
($CHBr_3$). These trace gases are a source of sulphate particles and halide radicals when oxidised in the
atmosphere, and have important roles as ozone catalysts in the troposphere and stratosphere (O'Dowd
*et al.*, 2002; Solomon *et al.*, 1994) and as cloud condensation nuclei (CCNs; Charlson *et al.,* 1987).
DMS is found globally in surface waters originating from the algal-produced precursor
dimethylsulphoniopropionate (DMSP, $C_5H_{10}O_2S$). Both DMS and DMSP are major routes of sulphur
and carbon flux through the marine microbial food web, and can provide up to 100% of the bacterial
(Simó *et al.*, 2009) and phytoplanktonic (Vila-Costa *et al.*, 2006a) sulphur demand. DMS is also a
volatile compound which readily passes through the marine boundary layer to the troposphere, where
oxidation results in a number of sulphur-containing particles important for atmospheric climate
feedbacks (Charlson *et al.*, 1987; Quinn and Bates, 2011); for this reason, any change in the production
of DMS may have significant implications for climate regulation. Several previous acidification
experiments have shown differing responses of both compounds (e.g. Avgoustidi *et al.,* 2012; Hopkins
*et al*., 2010; Webb *et al.*, 2015), while others have shown delayed or more rapid responses as a direct
effect of $CO_2$ (e.g. Archer *et al.*, 2013; Vogt *et al.*, 2008). Further, some laboratory incubations of
coastal microbial communities showed increased DMS production with increased $f$CO_2 (Hopkins and
Archer, 2014), but lower DMSP production. The combined picture arising from existing studies is that
the response of communities to $f$CO_2 perturbation is not predictable and requires further study.
Previous studies measuring DMS in the Baltic Sea measured concentrations up to 100 nmol $L^{-1}$ during
the summer bloom, making the Baltic Sea a significant source of DMS (Orlikowska and Schulz-Bull,

85  2009).

In surface waters, halocarbons such as methyl iodide ($CH_3I$), chloroiodomethane ($CH_2ClI$) and
bromoform ($CHBr_3$) are produced by biological and photochemical processes: many marine microbes
(for example cyanobacteria; Hughes *et al.*, 2011, diatoms; Manley and De La Cuesta, 1997 and
haptophytes; Scarratt and Moore, 1998) and macroalgae (e.g. brown-algal *Fucus* species; Chance *et*
*al.*, 2009 and red algae; Leedham *et al.*, 2013) utilise halides from seawater and emit a range of
organic and inorganic halogenated compounds. This production can lead to significant flux to the
marine boundary layer in the order of 10 Tg iodine-containing compounds ('iodocarbons'; O'Dowd *et*
*al.,* 2002) and 1 Tg bromine-containing compounds ('bromocarbons'; Goodwin *et al.*, 1997) into the



atmosphere. The effect of acidification on halocarbon concentrations has received limited attention,
but two acidification experiments measured lower concentrations of several iodocarbons while
bromocarbons were unaffected by $f$CO$_2$ up to 3000 µatm (Hopkins *et al.*, 2010; Webb, 2015), whereas
an additional mesocosm study did not elicit significant differences from any compound up to 1400
µatm $f$CO$_2$ (Hopkins *et al.*, 2013).
Measurements of the trace gases within the Baltic Sea are limited, with no prior study of DMSP
concentrations in the region. The Baltic Sea is the largest body of brackish water in the world, and
salinity ranges from 1 to 15. Furthermore, seasonal temperature variations of over 20 °C are common.
A permanent halocline at 50-80 m separates CO$_2$-rich, bottom waters from fresher, lower CO$_2$ surface
waters, and a summer thermocline at 20 m separates warmer surface waters from those below 4°C
(Janssen *et al.*, 1999). Upwelling of bottom waters from below the summer thermocline is a common
summer occurrence, replenishing the surface nutrients while simultaneously lowering surface
temperature and pH (Brutemark *et al.*, 2011). Baltic organisms are required to adapt to significant
variations in environmental conditions. The species assemblage in the Baltic Sea is different to those
studied during previous mesocosm experiments in the Arctic, North Sea and Korea (Brussaard *et al.*,
2013; Engel *et al.*, 2008; Kim *et al.*, 2010), and are largely unstudied in terms of their community trace
gas production during the summer bloom. Post-spring bloom (July-August), a low dissolved inorganic
nitrogen (DIN) to dissolved inorganic phosphorous (DIP) ratio combines with high temperatures and
light intensities to encourage the growth of heterocystous cyanobacteria, (Niemisto *et al.*, 1989;
Raateoja *et al.*, 2011), in preference to nitrate-dependent groups.
Here we report the concentrations of DMS, DMSP and halocarbons from the 2012 summer season
mesocosm experiment aimed to assess the impact of elevated $f$CO$_2$ on the microbial community and
trace gas production in the Baltic Sea. Our objective was to assess how changes in the microbial
community driven by changes in $f$CO$_2$ impacted DMS and halocarbon concentrations. It is anticipated
that any effect of CO$_2$ on the growth of different groups within the phytoplankton assemblage will
result in an associated change in trace gas concentrations measured in the mesocosms as $f$CO$_2$
increases, which can potentially be used to predict future halocarbon and sulphur emissions from the
Baltic Sea region.



**2   Methods**
**2.1   Mesocosm design and deployment**
Nine mesocosms were deployed on the 10th June 2012 (day $t$-10; days are numbered negative prior to
$CO_2$ addition and positive afterward) and moored near Tvärminne Zoological Station (59° 51.5' N, 23°
15.5' E) in Tvärminne Storfjärden in the Baltic Sea. Each mesocosm comprised a thermoplastic
polyurethane (TPU) enclosure of 17 m depth, containing approximately 54,000 L of seawater,
supported by an 8m tall floating frame capped with a polyvinyl hood. For full technical details of the
mesocosms see Czerny *et al.* (2013) and Riebesell *et al.* (2013). The mesocosm bags were filled by
lowering through the stratified water column until fully submerged, with the opening at both ends
covered by 3 mm mesh to exclude organisms larger than 3 mm such as fish. The mesocosms were then
left for 3 days ($t$-10 to $t$-7) with the mesh in position to allow exchange with the external water masses
and ensure the mesocosm contents were representative of the phytoplankton community in the
Storfjärden. On $t$-7 the bottom of the mesocosm was sealed with a sediment trap and the upper opening
was raised to approximately 1.5 m above the water surface. Stratification within the mesocosm bags
was broken up on $t$-5 by the use of compressed air for three and a half minutes to homogenise the
water column and ensure an even distribution of inorganic nutrients at all depths. Unlike in previous
experiments, there was no addition of inorganic nutrients to the mesocosms at any time during the
experiment; mean inorganic nitrate, inorganic phosphate and ammonium concentrations measured
across all mesocosms at the start of the experiment were 37.2 (± 18.8 s.d.) nmol $L^{-1}$, 323.9 (± 19.4 s.d.)
nmol $L^{-1}$ and 413.8 (± 319.5 s.d.) nmol $L^{-1}$ respectively.
To obtain mesocosms with different $f$CO$_2$, the carbonate chemistry of the mesocosms was altered by
the addition of different volumes of 50 μm filtered, $CO_2$-enriched Baltic Sea water (sourced from
outside the mesocosms), to each mesocosm over a four day period, with the first day of addition being
defined as day $t$0. Addition of the enriched $CO_2$ water was by the use of a bespoke dispersal apparatus
('Spider') lowered through the bags to ensure even distribution throughout the water column (further
details are in Riebesell *et al.* 2013). Measurements of salinity in the mesocosms throughout the
experiment determined that three of the mesocosms were not fully sealed, and had undergone
unquantifiable water exchange with the surrounding waters. These three mesocosms (M2, M4 and M9)
were excluded from the analysis. Two mesocosms were designated as controls (M1 and M5) and
received only filtered seawater via the Spider; four mesocosms received addition of $CO_2$-enriched
waters, with the range of target $f$CO$_2$ levels between 600 and 1650 μatm (M7, 600 μatm; M6, 950
μatm; M3, 1300 μatm; M8 1650 μatm). Mesocosms were randomly allocated a target $f$CO$_2$; a



noticeable decrease in $f$CO$_2$ was identified in the three highest $f$CO$_2$ mesocosms (M6, M3 and M8)
over the first half of the experiment, which required the addition of more CO$_2$ enriched water on $t$15 to
bring the $f$CO$_2$ back up to maximum concentrations (Fig. 1a; Paul *et al.*, 2015). A summary of the
$f$CO$_2$ in the mesocosms can be seen in Table 1. At the same time as this further CO$_2$ addition on $t$15,
the walls of the mesocosms were cleaned using a bespoke wiper apparatus (See Riebesell *et al.*, 2013
for more information), followed by weekly cleaning to remove aggregations on the film which would
block incoming light. Light measurements showed that over 95% of the photosynthetically active
radiation (PAR) was transmitted by the clean TPU and PVC materials with 100% absorbance of UV
light (Riebesell *et al.*, 2013). Samples for most parameters were collected from the mesocosms at the
same time every morning from $t$-3, and analysed daily or every other day.

## 2.2  Trace gas extraction and analysis

### 2.2.1  DMS and halocarbons

A depth-integrated water sampler (IWS, HYDRO-BIOS, Kiel, Germany) was used to sample the entire
17 m water column daily or alternative daily. As analysis of Chlorophyll-*a* (Chl-*a*) showed it to be
predominantly produced in the first 10 m of the water column; trace gas analysis was conducted on
only integrated samples collected from the surface 10 m, with all corresponding community parameter
analyses with the exception of pigment analysis performed also to this depth. Water samples for trace
gas analysis were taken from the first IWS from each mesocosm to minimise the disturbance and
bubble entrainment from taking multiple samples in the surface waters. As in Hughes *et al.* (2009),
samples were collected in 250 mL amber glass bottles in a laminar flow with minimal disturbance to
the water sample, using Tygon tubing from the outlet of the IWS. Bottles were rinsed twice before
being carefully filled from the bottom with minimal stirring, and allowed to overflow the volume of
the bottle approximately three times before sealing with a glass stopper to prevent bubble formation
and atmospheric contact. Samples were stored below 10°C in the dark for 2 hours prior to analysis.
Each day, a single sample was taken from each mesocosm, with two additional samples taken from
one randomly selected mesocosm to evaluate the precision of the analysis.
On return to the laboratory, 40 mL of water was injected into a purge and cryotrap system (Chuck *et*
*al.*, 2005), filtered through a 25 mm Whatman glass fibre filter (GF/F; GE Healthcare Life Sciences,
Little Chalfont, England) and purged with oxygen-free nitrogen (OFN) at 80 mL min$^{-1}$ for 10 minutes.
Each gas sample passed through a glass wool trap to remove particles and aerosols, before a dual
nafion counterflow drier (180 mL min$^{-1}$ OFN) removed water vapour from the gas stream. The gas
sample was trapped in a stainless steel loop held at -150 °C in the headspace of a liquid nitrogen-filled





dewar. The sample was injected by immersion of the sample loop in boiling water into an Agilent 6890
gas chromatograph equipped with a 60 m DB-VRX capillary column (0.32 mm ID, 1.8 μm film
thickness, Agilent J&W Ltd) according to the programme outlined by Hopkins *et al.* (2010). Analysis
was performed by an Agilent 5973 quadrupole mass spectrometer operated in electron ionisation,
single ion mode. Liquid standards of $CH_3I$, diiodomethane ($CH_2I_2$), $CH_2ClI$, iodoethane ($C_2H_5I$),
iodopropane ($C_3H_7I$), $CHBr_3$, dibromoethane ($CH_2Br_2$), dibromochloromethane ($CHBr_2Cl$),
bromoiodomethane ($CH_2BrI$) and DMS (Standards supplied by Sigma Aldrich Ltd, UK) were
gravimetrically prepared by dilution in HPLC-grade methanol (Table 2) and used for calibration. The
relative standard error was expressed as a percentage of the mean for the sample analysis, calculated
for each compound using triplicate analysis each day from a single mesocosm, and was <7% for all
compounds. GC-MS instrument drift was corrected by the use of a surrogate analyte standard in every
sample, comprising deuterated DMS ($D_6$-DMS), deuterated methyl iodide ($CD_3I$) and $^{13}C$
dibromoethane ($^{13}C_2H_4Br_2$) via the method described in Hughes *et al.* (2006) and Martino *et al.* (2005).
Five-point calibrations were performed weekly for each compound with the addition of the surrogate
analyte, with a single standard analysed daily to check for instrument drift; linear regression from
calibrations typically produced $r^2$>0.98. All samples measured within the mesocosms were within the
concentration ranges of the calibrations (Table 2).

### 2.2.2 DMSP

Samples for total DMSP ($DMSP_T$) were collected and stored for later analysis by the acidification
method of Curran *et al.* (1998). A 7 mL sub-sample was collected from the amber glass bottle into an 8
mL glass sample vial (Labhut, Churcham, UK), into which 0.35 μL of 50% $H_2SO_4$ was added, before
storage at ambient temperature. Particulate DMSP ($DMSP_P$) samples were prepared by the gravity
filtration of 20 mL of sample through a 47 mm GF/F in a glass filter unit, before careful removal and
folding of the GF/F into a 7 mL sample vial filled with 7 mL of Milli-Q water and 0.35 μL of $H_2SO_4$
before storage at ambient temperature. Samples were stored for approximately 8 weeks prior to
analysis. DMSP samples (total and particulate) were analysed on a PTFE purge and cryotrap system
using 2 mL of the sample purged with 1 mL of 10M NaOH for 5 minutes at 80 mL min$^{-1}$. The sample
gas stream passed through a glass wool trap and Nafion counterflow (Permapure) drier before being
trapped in a PTFE sample loop kept at -150 °C by suspension in the headspace of a liquid nitrogen-
filled dewar and controlled by feedback from a thermocouple. Immersion in boiling water rapidly re-
volatilised the sample for injection into a Shimadzu GC2010 gas chromatograph with a Varian
Chrompack CP-Sil-5CB column (30 m, 0.53 mm ID) and flame photometric detector (FPD). The GC
oven was operated isothermally at 60 °C which resulted in DMS eluting at 2.1 minutes. Liquid DMSP



standards were prepared and purged in the same manner as the sample to provide weekly calibrations
of the entire analytical system. Involvement in the 2013 AQA 12-23 international DMS analysis
proficiency test (National Measurement Institute of Australia, 2013) in February 2013 demonstrated
excellent agreement between our method of DMSP analysis and the mean from thirteen laboratories
measuring DMS using different methods, with a measurement error of 5%.
DMSP was not detected in any of the samples (total or particulate) collected and stored during the
experiment, and it was considered likely that this was due to an unresolved issue regarding acidifying
the samples for later DMSP analysis. It was considered unlikely that rates of bacterial DMSP turnover
through demethylation rather than through cleavage to produce DMS (Curson *et al.*, 2011) were
sufficiently high in the Baltic Sea to remove all detectable DMSP, yet still produce measureable DMS
concentrations. Also, rapid turnover of $DMSP_D$ in surface waters being the cause of low $DMSP_T$
concentrations does not explain the lack of intracellular particulate-phase DMSP. Although production
of DMS is possible from alternate sources, it is highly unlikely that there was a total absence of
DMSP-producing phytoplankton within the mesocosms or Baltic Sea surface waters around
Tvärminne; DMSP has been measured in surface waters of the Southern Baltic Sea at 22.2 nmol $L^{-1}$ in
2012, indicating that DMSP-producing species are present within the Baltic Sea (Cathleen Zindler,
GEOMAR, Pers. Comm.).
A previous study by del Valle *et al.* (2011) highlighted up to 94% loss of DMSP from acidified
samples of colonial *Phaeocystis globosa* culture, and field samples dominated by colonial *Phaeocystis*
*antarctica*. Despite filamentous, colonial cyanobacteria in the samples from Tvärminne mesocosms
potentially undergoing the same process, these species did not dominate the community at only 6.6%
of the total Chl-*a*, implying that the acidification method for DMSP fixation also failed for unicellular
phytoplankton species. This suggests that the acidification method is unreliable in the Baltic Sea, and
should be considered inadequate as the sole method of DMSP fixation in future experiments in the
region. The question of its applicability in other marine waters also needs further investigation.

**2.3  Measurement of community dynamics**
Water samples were collected from the 10m and 17mIWS on a daily basis and analysed for carbonate
chemistry, fluorometric Chl-*a*, phytoplankton pigments (17m IWS only) and cell abundance to analyse
the community structure and dynamics during the experiment. The carbonate system was analysed
through a suite of measurements (Paul *et al.*, 2015), including potentiometric titration for total
alkalinity (TA), infrared absorption for dissolved inorganic carbon (DIC) and spectrophotometric




determination for pH. For Chl-*a* analysis and pigment determination, 500 mL sub-samples were
filtered through a GF/F and stored frozen (-20 °C for two hours for Chl-*a* and -80 °C for up to 6
months  for pigments), before homogenisation in 90 % acetone with glass beads. After centrifuging
(10 minutes at 800 x g at 4 °C) the Chl-*a* concentrations were determined using a Turner AU-10
fluorometer  by the methods of Welschmeyer (1994), and the phytoplankton pigment concentrations
by reverse phase high performance liquid chromatography (WATERS HPLC with a Varian Microsorb-
MV 100-3 C8 column) as described by Barlow *et al.* (1997). Phytoplankton community composition
was determined by the use of the CHEMTAX algorithm to convert the concentrations of marker
pigments to Chl-*a* equivalents (Mackey *et al.*, 1996; Schulz *et al.*, 2013). Microbes were enumerated
using a Becton Dickinson FACSCalibur flow cytometer (FCM) equipped with a 488 nm argon laser
(Crawfurd *et al.*, 2015) and counts of phytoplankton cells >20 μm were made on concentrated (50 mL)
sample water, fixed with acidic Lugol's iodine solution with an inverted microscope. Filamentous
cyanobacteria were counted in 50 μm length units.
**2.4   Statistical Analysis**
All statistical analysis was performed using Minitab V16. In analysis of the measurements between
mesocosms, one-way ANOVA was used with Tukey's post-hoc analysis test to determine the effect of
different $f$CO$_2$ on concentrations measured in the mesocosms and the Baltic Sea. Spearman's Rank
Correlation Coefficients were calculated to compare the relationships between trace gas
concentrations, $f$CO$_2$, and a number of biological parameters, and the resulting $\rho$-values for each
correlation are given in Supplementary table S1 for the mesocosms and S2 for the Baltic Sea data.

**3   Results and Discussion**
**3.1   Biogeochemical changes within the mesocosms**
The mesocosm experiment was split into three phases based on the temporal variation in Chl-*a* (Fig. 2;
Paul *et al.*, 2015) evaluated after the experiment was completed:
• Phase 0 (days *t*-5 to *t*0) – pre-CO$_2$ addition
• Phase I (days *t*1 to *t*16) – 'productive phase'
• Phase II (days *t*17 to *t*30) – temperature induced autotrophic decline.





### 3.1.1 Physical Parameters

$f$CO$_2$ decreased over Phase I in the three highest $f$CO$_2$ mesocosms, mainly through air-sea gas
exchange and carbon fixation by phytoplankton (Fig. 1a). All mesocosms still showed distinct
differences in $f$CO$_2$ levels throughout the experiment (Table 1), and there was no overlap of mesocosm
$f$CO$_2$ values on any given day, save for the two controls (M1 and M5). The control mesocosm $f$CO$_2$
increased through Phase I of the experiment, likely as a result of undersaturation of the water column
encouraging dissolution of atmospheric CO$_2$ (Paul *et al.*, 2015). Salinity in the mesocosms remained
constant throughout the experiment at 5.70 ± 0.004, and showed no variation with depth. It remained
similar to salinity in the Baltic Sea surrounding the mesocosms, which was 5.74 ± 0.14. Water
temperature varied from a low of 8.6 ± 0.4 °C during Phase 0 to a high of 15.9 ± 2.2 °C measured on
day $t$16, before decreasing once again (Fig. 1b).
Summertime upwelling events are common and well described (Gidhagen, 1987; Lehmann and
Myrberg, 2008), and induce a significant temperature decrease in surface waters; such an event
appears to have commenced around $t$16, as indicated by significantly decreasing temperatures inside
and out of the mesocosms (Fig. 1b) and increased salinity in the Baltic Sea from 5.5 to 6.1 over the
following 15 days to the end of the experiment. Due to the enclosed nature of the mesocosms, the
upwelling affected only the temperature and not pH, $f$CO$_2$ or the microbial community. However, the
temperature decrease after $t$16 was likely to have had a significant effect on phytoplankton growth,
explaining the lower Chl-$a$ in Phase II.

### 3.1.2 Community Dynamics

Mixing of the mesocosms after closure prior to $t$-3 did not trigger a notable increase in Chl-$a$ in Phase
0; in previous mesocosm experiments, mixing redistributed nutrients from the deeper stratified layers
throughout the water column. During Phase I, light availability, combined with increasing water
temperatures favoured the growth of phytoplankton in all mesocosms (Paul *et al.* 2015), and was
unlikely to be a direct result of the CO$_2$ enrichment. Mean Chl-$a$ during Phase I was 1.98 (± 0.29) μg
L$^{-1}$ from all mesocosms, decreasing to 1.44 (± 0.46) μg L$^{-1}$ in Phase II: this decrease was attributed to a
temperature induced decreased in phytoplankton growth rates and higher grazing rates as a result of
higher zooplankton reproduction rates during Phase I (Lischka *et al.*, 2015; Paul *et al.*, 2015).
Mesocosm Chl-$a$ decreased until the end of the experiment on $t$31.
The largest contributors to Chl-$a$ in the mesocosms during the summer of 2012 were the chlorophytes
and cryptophytes, with up to 40% and 21% contributions to the Chl-$a$ respectively (Table 3; Paul *et al.*,
2015). Significant long-term differences in abundance between mesocosms developed as a result of



elevated $f$CO$_2$ in only two groups: picoeukaryotes I showed higher abundance at high $f$CO$_2$ (F=8.2,
p<0.01; Crawfurd *et al.*, 2016 and Supplementary Fig. S2), as seen in previous mesocosm experiments
(Brussaard *et al.*, 2013; Newbold *et al.*, 2012) and picoeukaryotes III the opposite trend (F=19.6,
p<0.01; Crawfurd *et al.* this issue). Temporal variation in phytoplankton abundance was similar
between all mesocosms (Supplementary Fig. S1 and S2).
Diazotrophic, filamentous cyanobacterial blooms in the Baltic Sea are an annual event in summer
(Finni *et al.*, 2001), and single-celled cyanobacteria have been found to comprise as much as 80% of
the cyanobacterial biomass and 50% of the total primary production during the summer in the Baltic
Sea (Stal *et al.*, 2003). However, CHEMTAX analysis identified cyanobacteria as contributing less
than 10% of the total Chl-*a* in the mesocosms (Crawfurd *et al.*, 2015; Paul *et al.*, 2015). These
observations were backed up by satellite observations showing reduced cyanobacterial abundance
throughout the Baltic Sea in 2012 compared to previous and later years (Oberg, 2013). It was proposed
that environmental conditions of limited light availability and lower surface water temperatures during
the summer of 2012 were sub-optimal for triggering a filamentous cyanobacteria bloom (Wasmund,

326  1997).


### 3.2  DMS and DMSP

### 3.2.1  Mesocosm DMS

A significant 34% reduction in DMS concentrations was detected in the high $f$CO$_2$ treatments during
Phase II compared to the ambient $f$CO$_2$ mesocosms (F=31.7, p<0.01). Mean DMS concentrations of
5.0 ($\pm$ 0.8; range 3.5 – 6.8) nmol L$^{-1}$ in the ambient treatments compared to 3.3 ($\pm$ 0.3; range 2.9 – 3.9)
nmol L$^{-1}$ in the 1333 and 1075 µatm mesocosms (Fig. 3a). The primary differences identified were
apparent from the start of Phase II on $t$17, after which maximum concentrations were observed in the
ambient mesocosms on $t$21. The relationship between DMS and increasing $f$CO$_2$ during Phase II was
found to be linear (Fig. 3b), a finding also identified in previous mesocosm experiments (Archer *et al.*,
2013; Webb *et al.*, 2015). Furthermore, increases in DMS concentrations under high $f$CO$_2$ were
delayed by three days relative to the ambient and medium $f$CO$_2$ treatments, a situation which has been
observed in a previous mesocosm experiment. This was attributed to small-scale shifts in community
composition and succession which could not be identified with only a once-daily measurement regime
(Vogt *et al.*, 2008). DMS measured in all mesocosms fell within the range 2.7 to 6.8 nmol L$^{-1}$ across
the course of the experiment. During Phase I, no difference was identified in DMS concentrations



between $f$CO$_2$ treatments with the mean of all mesocosms 3.1 (± 0.2) nmol L$^{-1}$. Concentrations in all
mesocosms gradually declined from $t$21 until the end of DMS measurements on $t$31. DMS
concentrations measured in the mesocosms and Baltic Sea were comparable to those measured in
temperate coastal conditions in the North Sea (Turner *et al.*, 1988), the Mauritanian upwelling
(Franklin *et al.*, 2009; Zindler *et al.*, 2012) and South Pacific (Lee *et al.*, 2010).
Although the majority of DMS production is presumed to be from DMSP, an alternative production
route for DMS is available through the methylation of methanethiol (Drotar *et al.*, 1987; Kiene and
Hines, 1995; Stets *et al.*, 2004) predominantly identified in anaerobic environments such as freshwater
lake sediments (Lomans *et al.*, 1997), saltmarsh sediments (Kiene and Visscher, 1987) and microbial
mats (Visscher *et al.*, 2003; Zinder *et al.*, 1977). However, recent studies have identified this pathway
of DMS production from *Pseudomonas deceptionensis* in an aerobic environment (Carrión *et al.*,
2015), where *P. deceptionensis* was unable to synthesis or catabolise DMSP, but was able to
enzymatically mediate DMS production from methanethiol (MeSH). The same enzyme has also been
identified in a wide range of other bacterial taxa, including the cyanobacterial *Pseudanabaena*, which
was identified in the Baltic Sea during this and previous investigations (Stuhr, pers. comm.; Kangro *et*
*al.*, 2007; Nausch *et al.*, 2009). Correlations between DMS and the cyanobacterial equivalent Chl-*a*
($\rho$=0.42, p<0.01) indicate that the methylation pathway may be a potential source of DMS within the
Baltic Sea community. In addition to the methylation pathway, DMS production has been identified
from S-methylmethionine (Bentley and Chasteen, 2004), as well as from the reduction of
dimethylsulphoxide (DMSO) in both surface and deep waters by bacterial metabolism (Hatton *et al.*,
2004). As these compounds were not measured in the mesocosms, it is impossible to determine if they
were significant sources of DMS.
### 3.2.2 DMS and Community Interactions
Throughout Phase I, DMS showed no correlation with any measured variables of biological activity or
cell abundance, and was unaffected by elevated $f$CO$_2$, indicating DMS net production was not directly
related to the perturbation of the system and associated cellular stress (Sunda *et al.*, 2002). During
Phase II, DMS was negatively correlated with Chl-*a* in the ambient and medium $f$CO$_2$ mesocosms ($\rho$=-
0.60, p<0.01). During Phase II, a significant correlation was seen between DMS and single-celled
cyanobacteria identified as *Synechococcus* ($\rho$=0.53, p<0.01; Crawfurd *et al.* 2016 and supplementary
table S1) and picoeukaryotes III ($\rho$=0.75, p<0.01). The peak in DMS concentrations is unlikely to be a
delayed response to the increased Chl-*a* on $t$16.



In previous mesocosm experiments (Archer *et al.*, 2013; Hopkins *et al.*, 2010; Webb *et al.*, 2015),
DMS has shown poor correlations with many of the indicators of primary production and
phytoplankton abundance, as well as showing the same trend of decreased concentrations in high $f$CO$_2$
mesocosms compared to ambient. DMS production is often uncoupled from measurements of primary
production in open waters (Lana *et al.*, 2012), and also often from production of its precursor DMSP
(Archer *et al.*, 2009).. DMS and DMSP are important sources of sulphur and carbon in the microbial
food web for both bacteria and algae (Simó *et al.*, 2002, 2009), and since microbial turnover of DMSP
and DMS play a significant role in net DMS production, it is unsurprising that DMS concentrations
have shown poor correlation with DMSP-producing phytoplankton groups in past experiments and
open waters.
DMS concentrations have been reported lower under conditions of elevated $f$CO$_2$ compared to ambient
controls, in both mesocosm experiments (Table 4) and phytoplankton monocultures (Arnold *et al.*,
2013; Avgoustidi *et al.*, 2012). However, these experiments limit our ability to generalise the response
of algal production of DMS and DMSP in all situations due to the characteristic community dynamics
of each experiment in specific geographical areas and temporal periods. Previous experiments in the
temperate Raunefjord of Bergen, Norway, showed lower abundance of DMSP-producing algal species,
and subsequently DMSP-dependent DMS concentrations (Avgoustidi *et al.*, 2012; Hopkins *et al.*,
2010; Vogt *et al.*, 2008; Webb *et al.*, 2015). In contrast mesocosm experiments in the Arctic and Korea
have shown increased abundance of DMSP producers (Archer *et al.*, 2013; Kim *et al.*, 2010) but lower
DMS concentrations, while incubation experiments by Hopkins and Archer (2014) showed lower
DMSP production but higher DMS concentrations at high $f$CO$_2$. However, in all previous experiments
with DMSP as the primary precursor of DMS, elevated $f$CO$_2$ had a less marked effect on measured
DMSP concentrations than on measured DMS concentrations. Hopkins *et al.* (2010) suggested that
'the perturbation of the system has a greater effect on the processes that control the conversion of
DMSP to DMS rather than the initial production of DMSP itself'. This is relevant even for the current
experiment, where DMSP was not identified, since processes controlling DMS concentrations were
likely more affected by the change in $f$CO$_2$ than the production of precursors.
Previous mesocosm experiments have suggested significant links between increased bacterial
production through greater availability of organic substrates at high $f$CO$_2$ (Engel *et al.*, 2013; Piontek
*et al.*, 2013). Further, Endres *et al.* (2014) identified significant enhanced enzymatic hydrolysis of
organic matter with increasing $f$CO$_2$, with higher bacterial abundance. Higher bacterial abundance will
likely result in greater bacterial demand for sulphur, and therefore greater consumption of DMS and
conversion to DMSO. This was suggested as a significant sink for DMS in a previous experiment



(Webb *et al.*, 2015), but during the present experiment, both bacterial abundance and bacterial
production were lower at high $f$CO$_2$ (Hornick *et al.*, 2015). However, as it has been proposed that only
specialist bacterial groups are DMS consumers (Vila-Costa *et al.*, 2006b), and there is no
determination of the DMS consumption characteristics of the bacterial community in the Baltic Sea,
this is still a potential stimulated DMS loss pathway at high $f$CO$_2$. *Synechococcus* has been identified
as a DMS consumer in the open ocean, but abundance of this group was negatively correlated with
$f$CO$_2$, implying that DMS consumption by this group would have been lower as $f$CO$_2$ increased.

### 3.3  Iodocarbons in the mesocosms and relationships with community composition

Elevated $f$CO$_2$ did not affect the concentration of iodocarbons in the mesocosms significantly at any
time during the experiment, which is in agreement with the findings of Hopkins *et al.* (2013) in the
Arctic, but in contrast to Hopkins *et al.* (2010) and Webb (2015), where iodocarbons were measured
significantly lower under elevated $f$CO$_2$ (Table 4). Concentrations of all iodocarbons measured in the
mesocosms and the Baltic Sea fall within the range of those measured previously in the region (Table
5). Mesocosm concentrations of CH$_3$I (Fig. 4a) and C$_2$H$_5$I (Fig. 4b) showed concentration ranges of
2.91 to 6.25 and 0.23 to 0.76 pmol L$^{-1}$ respectively. CH$_3$I showed a slight increase in all mesocosms
during Phase I, peaking on $t$16 which corresponded with higher Chl-*a* concentrations, and correlated
throughout the entire experiment with picoeukaryote groups II ($\rho$=0.59, p<0.01) and III ($\rho$=0.23,
p<0.01; Crawfurd *et al.* this issue) and nanoeukaryotes I ($\rho$=0.37, p<0.01). Significant differences
identified between mesocosms for CH$_3$I were unrelated to elevated $f$CO$_2$ (F=3.1, p<0.05), but
concentrations were on average 15% higher in Phase II than Phase I. C$_2$H$_5$I decreased slightly during
Phases I and II, although concentrations of this halocarbon were close to its detection limit (0.2 pmol
L$^{-1}$), remaining below 1 pmol L$^{-1}$ at all times. As this compound showed no significant effect of
elevated $f$CO$_2$, and was identified by Orlikowska and Schulz-Bull (2009) as having extremely low
concentrations in the Baltic Sea (Table 5), it will not be discussed further.
No correlation was found between CH$_3$I and Chl-*a* at any phase, and the only correlation of any
phytoplankton grouping was with nanoeukaryotes II ($\rho$=0.88, p<0.01; Crawfurd *et al.*, 2015). These
CH$_3$I concentrations compare well to the 7.5 pmol L$^{-1}$ measured by Karlsson *et al.* (2008) during a
cyanobacterial bloom in the Baltic Sea (Table 5), and the summer maximum of 16 pmol L$^{-1}$ identified
by Orlikowska and Schulz-Bull (2009).
Karlsson *et al.* (2008) showed Baltic Sea halocarbon production occurring predominately during
daylight hours, with concentrations at night decreasing by 70% compared to late afternoon. Light
dependent production of CH$_3$I has been shown to take place through abiotic processes, including



radical recombination of $CH_3$ and I (Moore and Zafiriou, 1994). However since samples were
integrated over the surface 10m of the water column, it was impossible to determine if photochemistry
was affecting iodocarbon concentrations near the surface where some UV light was able to pass
between the top of the mesocosm film material and the cover. For the same reason, photodegradation
of halocarbons (Zika *et al.*, 1984) within the mesocosms was also likely to have been significantly
restricted. Thus, as photochemical production was expected to be minimal, biogenic production was
likely to have been the dominant source of these compounds. Karlsson *et al.* (2008) identified
*Pseudanabaena* as a key producer of $CH_3I$ in the Baltic Sea. However the abundance of
*Pseudanabaena* was highest during Phase I of the experiment (A. Stuhr, Pers. Comm.) when $CH_3I$
concentrations were lower, and as discussed previously, the abundance of these species constituted
only a very small proportion of the community. Previous investigations in the laboratory have
identified diatoms as significant producers of $CH_3I$ (Hughes *et al.*, 2013; Manley and De La Cuesta,
1997), and the low, steady-state abundance of the diatom populations in the mesocosms could have
produced the same relatively steady-state trends in the iodocarbon concentrations.
Measured in the range 57.2 – 202.2 pmol $L^{-1}$ in the mesocosms, $CH_2I_2$ (Fig. 4c) showed the clearest
increase in concentration during Phase II, when it peaked on $t21$ in all mesocosms, with a maximum of
202.2 pmol $L^{-1}$ in M5 (348 µatm). During Phase II, concentrations of $CH_2I_2$ were 57% higher than
Phase I, and were therefore negatively correlated with Chl-*a*. The peak on $t21$ corresponds with the
peak identified in DMS on $t21$, and concentrations through all three phases correlate with
picoeukaryotes II ($\rho$=0.62, p<0.01) and III ($\rho$=0.47, p<0.01) and nanoeukaryotes I ($\rho$=0.88, p<0.01;
Crawfurd *et al.,* 2015). $CH_2ClI$ (Fig. 4d) showed no peaks during either Phase I or Phase II, remaining
within the range 3.81 to 8.03 pmol $L^{-1}$, and again correlated with picoeukaryotes groups II ($\rho$=0.34,
p<0.01) and III ($\rho$=0.38, p<0.01). These results may suggest that these groups possessed halo-
peroxidase enzymes able to oxidise $I^-$, most likely as an anti-oxidant mechanism within the cell to
remove $H_2O_2$ (Butler and Carter-Franklin, 2004; Pedersen *et al.*, 1996; Theiler *et al.*, 1978). However,
given the lack of response of these compounds to elevated $f$CO$_2$ (F=1.7, p<0.01), it is unlikely that
production was increased in relation to elevated $f$CO$_2$. Production of all iodocarbons increased during
Phase II when total Chl-*a* decreased, particularly after the walls of the mesocosms were cleaned for the
first time, releasing significant volumes of organic aggregates into the water column. Aggregates have
been suggested as a source of $CH_3I$ and $C_2H_5I$ (Hughes *et al.*, 2008), likely through the alkylation of
inorganic iodide (Urhahn and Ballschmiter, 1998) or through the breakdown of organic matter by
microbial activity to supply the precursors required for iodocarbon production (Smith *et al.*, 1992).
Hughes *et al.* (2008) did not identify this route as a pathway for $CH_2I_2$ or $CH_2ClI$ production, but



Carpenter *et al.* (2005) suggested a production pathway for these compounds through the reaction of HOI with aggregated organic materials.

### 3.4 Bromocarbons in the mesocosms and the relationships with community composition

No effect of elevated $f\mathrm{CO_2}$ was identified for any of the three bromocarbons, which compared with the findings from previous mesocosms where bromocarbons were studied (Hopkins *et al.*, 2010, 2013; Webb, 2015; Table 4). Measured concentrations were comparable to those of Orlikowska and Schulz-Bull (2009) and Karlsson *et al.* (2008) measured in the Southern part of the Baltic Sea (Table 3). The concentrations of $CHBr_3$, $CH_2Br_2$ and $CHBr_2Cl$ showed no major peaks of production in the mesocosms. $CHBr_3$ (Fig. 5a) decreased rapidly in all mesocosms over Phase 0 from a maximum measured concentration of 147.5 pmol $L^{-1}$ in M1 (mean of 138.3 pmol $L^{-1}$ in all mesocosms) to a mean of 85.7 (±8.2 s.d.) pmol $L^{-1}$ in all mesocosms for the period $t0$ to $t31$ (Phases I and II). The steady-state $CHBr_3$ concentrations indicated a production source, however there was no clear correlation with any measured algal groups. $CH_2Br_2$ concentrations (Fig. 5b) decreased steadily in all mesocosms from $t$-3 through to $t31$, over the range 4.0 to 7.7 pmol $L^{-1}$, and $CHBr_2Cl$ followed a similar trend in the range 1.7 to 4.7 pmol $L^{-1}$ (Fig. 5c). Of the three bromocarbons, only $CH_2Br_2$ showed correlation with total Chl-*a* ($\rho$=0.52, p<0.01), and with cryptophyte ($\rho$=0.86, p<0.01) and dinoflagellate ($\rho$=0.65, p<0.01) derived Chl-*a*. Concentrations of $CH_2BrI$ were below detection limit for the entire experiment.

$CH_2Br_2$ showed positive correlation with Chl-*a* ($\rho$=0.52, p<0.01), nanoeukaryotes II ($\rho$=0.34, p<0.01) and cryptophytes ($\rho$=0.86, p<0.01; see supplementary material), whereas $CHBr_3$ and $CHBr_2Cl$ showed very weak or no correlation with any indicators of primary production. Schall *et al.* (1997) have proposed that $CHBr_2Cl$ is produced in seawater by the nucleophilic substitution of bromide by chloride in $CHBr_3$, which given the steady-state concentrations of $CHBr_3$ would explain the similar distribution of $CHBr2Cl$ concentrations. Production of all three bromocarbons was identified from large-size cyanobacteria such as *Aphanizomenon flos-aquae* by Karlsson *et al.* (2008), and in addition, significant correlations were found in the Arabian Sea between the abundance of the cyanobacterium *Trichodesmium* and several bromocarbons (Roy *et al.*, 2011), and the low abundance of such bacteria in the mesocosms would explain the low variation in bromocarbon concentrations through the experiment.

Halocarbon loss processes such as nucleophilic substitution (Moore, 2006), hydrolysis (Elliott and Rowland, 1995), sea-air exchange and microbial degradation are suggested as of greater importance than production of these compounds by specific algal groups, particularly given the relatively low



growth rates and total Chl-$a$. Hughes $et$ $al.$ (2013) identified bacterial inhibition of $CHBr_3$ production
in laboratory cultures of *Thalassiosira* diatoms, but that it was not subject to bacterial breakdown;
which could explain the relative steady state of $CHBr_3$ concentrations in the mesocosms. In contrast,
significant bacterial degradation of $CH_2Br_2$ in the same experiments could explain the steady decrease
in $CH_2Br_2$ concentrations seen in the mesocosms. Bacterial oxidation was also identified by Goodwin
$et$ $al.$ (1998) as a significant sink for $CH_2Br_2$. As discussed for the iodocarbons, photolysis was
unlikely due to the UV absorption of the mesocosm film, and limited UV exposure of the surface
waters within the mesocosm due to the mesocosm cover. The ratio of $CH_2Br_2$ to $CHBr_3$ was also
unaffected by increased $f$CO$_2$, staying within the range 0.04 to 0.08. This range in ratios is consistent
with that calculated by Hughes $et$ $al.$ (2009) in the surface waters of an Antarctic depth profile, and
attributed to higher sea-air flux of $CHBr_3$ than $CH_2Br_2$ due to a greater concentrations gradient, despite
the similar transfer velocities of the two compounds (Quack $et$ $al.$, 2007). Using cluster analysis in a
time-series in the Baltic Sea, Orlikowska and Schulz-Bull (2009) identified both these compounds as
originating from different sources and different pathways of production.
Macroalgal production would not have influenced the mesocosm concentrations due to the isolation
from the coastal environment, however the higher bromocarbon concentrations identified in the
mesocosms during Phase 0 may have originated from macroalgal sources (Klick, 1992; Leedham $et$
$al$., 2013; Moore and Tokarczyk, 1993) prior to mesocosm closure, with concentrations decreasing
through turnover and transfer to the atmosphere.

### 3.5   Natural variations in Baltic Sea $f$CO$_2$ and the effect on biogenic trace gases


### 3.5.1   Physical variation and community dynamics


Baltic Sea deep waters have high $f$CO$_2$ and subsequently lower pH (Schneider $et$ $al.$, 2002), and the
influx to the surface waters surrounding the mesocosms resulted in $f$CO$_2$ increasing to 725 µatm on
$t$31, close to the average $f$CO$_2$ of the third highest mesocosm (M6: 868 µatm). These conditions imply
that pelagic communities in the Baltic Sea are regularly exposed to rapid changes in $f$CO$_2$ and the
associated pH, as well as having communities associated with the elevated $f$CO$_2$ conditions.
Chl-$a$ followed the pattern of the mesocosms until $t$4, after which concentrations were significantly
higher than any mesocosm, peaking at 6.48 µg L$^{-1}$ on $t$16, corresponding to the maximum Chl-$a$ peak
in the mesocosms and the maximum peak of temperature. As upwelled water intruded into the surface
waters, the surface Chl-$a$ was diluted with low Chl-$a$ deep water: Chl-$a$ in the surface 10m decreased





from around $t16$ at the start of the upwelling until $t31$ when concentrations were once again equivalent
to those found in the mesocosms at 1.30 µg L$^{-1}$. In addition there was potential introduction of different
algal groups to the surface, but chlorophytes and crytophytes were the major contributors to the Chl-a
in the Baltic Sea, as in the mesocosms. Cyanobacteria contributed less than 2% of the total Chl-a in the
Baltic Sea (Crawfurd *et al*., 2015; Paul *et al.*, 2015).
Temporal community dynamics in the Baltic Sea were very different to that in the mesocosms across
the experiment, with euglenophytes, chlorophytes, diatoms and prasinophytes all showing distinct
peaks at the start of Phase II, with these same peaks identified in the nanoeukaryotes I and II, and
picoeukaryotes II (Crawfurd *et al.*, 2016; Paul *et al.*, 2015; Supplementary Figs. S1 and S2). The
decrease in abundance of many groups during Phase II was attributed to the decrease in temperature
and dilution with low-abundance deep waters.
### 3.5.2 DMS in the Baltic Sea
The input of upwelled water into the region mid-way through the experiment significantly altered the
biogeochemical properties of the waters surrounding the mesocosms, and as a result it is inappropriate
to directly compare the community structure and trace gas production of the Baltic Sea and the
mesocosms. The Baltic Sea samples gave a mean DMS concentration of 4.6 ± 2.6 nmol L$^{-1}$.but peaked
at 11.2 nmol L$^{-1}$ on $t16$, and were within the range of previous measurements for the region (Table 5).
Strong correlations were seen between DMS and Chl-*a* ($\rho$=0.84, p<0.01), with the ratio of DMS: Chl-*a*
at 1.6 (± 0.3) nmol µg$^{-1}$. Other strong correlations were seen with euglenophytes ($\rho$=0.89, p<0.01),
dinoflagellates ($\rho$=0.61, p<0.05) and nanoeukaryotes II ($\rho$=0.88, p<0.01), but no correlation was found
between DMS and cyanobacterial abundance, or with picoeukaryotes III which was identified in the
mesocosms, suggesting that DMS had a different origin in the Baltic Sea community than in the
mesocosms. Once again, there was no DMSP detected in the samples.
As $CO_2$ levels increased during Phase II, the DMS concentration measured in the Baltic Sea decreased,
from the peak on $t16$ to the lowest recorded sample of the entire experiment at 1.85 nmol L$^{-1}$. As with
Chl-*a*, DMS concentrations in the surface of the Baltic Sea may have been diluted with low-DMS deep
water, however, the inverse relationship of DMS with $CO_2$ shown in the mesocosms may suggest that
this decrease in DMS is attributed to the increase in $CO_2$ levels. Bacterial abundance was similar in the
Baltic Sea as in the mesocosms (Hornick *et al.*, 2015), however the injection of high $CO_2$ water may
have stimulated bacterial consumption of DMS during the upwelling, which combined with the
dilution of DMS-rich surface water could have resulted in the rapid decrease in DMS concentrations.
As no discernible decrease in total bacterial abundance was identified during the upwelling, it is also



possible that the upwelled water contained a different microbial community, and may potentially have
introduced a higher abundance of DMS-consuming microbes. No breakdown of bacterial distributions
was available with which to test this hypothesis.
### 3.5.3 Halocarbon concentrations in the Baltic Sea
Outside the mesocosms in the Baltic Sea, $CH_3I$ was measured at a maximum concentration of 8.65
pmol $L^{-1}$, during Phase II, and showed limited effect of the upwelling event. Both $CH_2I_2$ and $CH_2ClI$
showed higher concentrations in the Baltic Sea samples than the mesocosms ($CH_2I_2$: 373.9 pmol $L^{-1}$
and $CH_2ClI$: 18.1 pmol $L^{-1}$), and were correlated with the euglenophytes ($CH_2I_2$; $\rho$=0.63, p<0.05 and
$CH_2ClI$; $\rho$=0.68, p<0.01) and nanoeukaryotes II ($CH_2I_2$; $\rho$=0.53, p<0.01 and $CH_2ClI$; $\rho$=0.58, p<0.01),
but no correlation with Chl-$a$. Both polyiodinated compounds showed correlation with picoeukaryote
groups II and III, indicating that production was not limited to a single source. These concentrations of
$CH_2I_2$ and $CH_2ClI$ compared well to those measured over a macroalgal bed in the higher saline waters
of the Kattegat by Klick and Abrahamsson (1992), suggesting that macroalgae were a significant
iodocarbon source in the Baltic Sea.
As with the iodocarbons, the Baltic Sea showed significantly higher concentrations of $CHBr_3$ (F=28.1,
p<0.01), $CH_2Br_2$ (F=208.8, p<0.01) and $CHBr_2Cl$ (F=23.5, p<0.01) than the mesocosms, with
maximum concentrations 191.6 pmol $L^{-1}$, 10.0 pmol $L^{-1}$ and 5.0 pmol $L^{-1}$ respectively. In the Baltic
Sea, only $CHBr_3$ was correlated with Chl-$a$ ($\rho$=0.65, p<0.05), cyanobacteria ($\rho$=0.61, p<0.01; Paul $et$
$al.$, 2015) and nanoeukaryotes II ($\rho$=0.56, p<0.01; Crawfurd $et$ $al.,$ 2015), with the other two
bromocarbons showing little to no correlations with any parameter of community activity. Production
of bromocarbons from macroalgal sources (Laturnus $et$ $al.$, 2000; Leedham $et$ $al.$, 2013; Manley $et$ $al.$,
1992) was likely a significant contributor to the concentrations detected in the Baltic Sea; over the
macroalgal beds in the Kattegat, Klick (1992) measured concentrations an order of magnitude higher
than seen in this experiment for $CH_2Br_2$ and $CHBr_2Cl$.

## 4   The Baltic Sea as a natural analogue to future ocean acidification?
Mesocosm experiments are a highly valuable tool in assessing the potential impacts of elevated $CO_2$
on complex marine communities, however they are limited in that the rapid change in $f$$CO_2$
experienced by the community may not be representative of changes in the future ocean (Passow and
Riebesell, 2005). This inherent problem with mesocosm experiments can be overcome through using
naturally low pH/ high $CO_2$ areas such as upwelling regions or vent sites (Hall-Spencer $et$ $al.,$ 2008),
which can give an insight into populations already living and adapted to high $CO_2$ regimes by exposure




over timescales measured in years. This mesocosm experiment was performed at such a location with a
relatively low $f$CO$_2$ excursion compared to some sites (800 μatm compared to >2000 μatm; Hall-
Spencer et al., 2008), and it was clear through the minimal variation in Chl-*a* between all mesocosms
that the community was relatively unaffected by elevated $f$CO$_2$, although variation could be identified
in some phytoplankton groups and some shifts in community composition. The upwelling event
occurring mid-way through our experiment allowed comparison of the mesocosm findings with a
natural analogue of the system, as well as showing the extent to which the system perturbation can
occur (up to 800 μatm). However, it is very difficult to determine where and when an upwelling will
occur, and therefore hard to utilise these events as natural high CO$_2$ analogues.
In this paper, we described the temporal changes in concentrations of DMS and halocarbons in natural
Baltic phytoplankton communities exposed to elevated $f$CO$_2$ treatments. In contrast to the halocarbons,
concentrations of DMS were significantly lower in the highest $f$CO$_2$ treatments compared to the
control. Despite very different physicochemical and biological characteristics of the Baltic Sea (e.g.
salinity, community composition and nutrient concentrations), this is a very similar outcome to that
seen in several other high $f$CO$_2$ experiments. The Baltic Sea trace gas samples give a good record of
trace gas production during the injection of high $f$CO$_2$ deep water into the surface community during
upwelling events. For the concentrations of halocarbons, no response was shown to the upwelling
event in the Baltic Sea, which may indicate that emissions of organic iodine and bromine are unlikely
to change with future acidification of the Baltic Sea. However, production of organic sulphur within
the Baltic Sea region is likely to decrease with an acidified future ocean scenario, despite the possible
acclimation of the microbial community to elevated $f$CO$_2$. This will potentially impact the flux of
DMS to the atmosphere over Northern Europe, and could have significant impacts on the local climate
through the reduction of atmospheric sulphur aerosols. Data from a previous mesocosm experiment
has been used to estimate future global changes in DMS production, and predicted that global warming
would be amplified (Six *et al.,* 2013); utilising the data from this experiment combined with those of
other mesocosm, field and laboratory experiments and associated modelling provide the basis for a
better understanding of the future changes in global DMS production and their climatic impacts.



**Acknowledgements**
The Tvärminne 2012 mesocosm experiment was part of the SOPRAN II (Surface Ocean Processes in
the Anthropocene) Programme (FKZ 03F0611) and BIOACID II (Biological Impacts of Ocean
Acidification) project (FKZ 03F06550), funded by the German Ministry for Education and Research
(BMBF) and led by the GEOMAR Helmholtz Centre for Ocean Research Kiel, Germany. The authors
thank all participants in the SOPRAN Tvärminne experiment for their assistance, including A. Ludwig
for logistical support, the diving team, and the staff of Tvärminne Zoological Research Station for
hosting the experiment. We also acknowledge the captain and crew of RV *ALKOR* (**AL394 and**
**AL397)** for their work transporting, deploying and recovering the mesocosms.
This work was funded by a UK Natural Environment Research Council Directed Research Studentship
(NE/H025588/1) through the UK Ocean Acidification Research Programme, with CASE funding from
Plymouth Marine Laboratory. Additional funding was supplied by the EU Seventh Framework
Program (FP7/2007-2013) MESOAQUA (EC Contract No. 228224).



Archer, S. D., Cummings, D., Llewellyn, C. and Fishwick, J.: Phytoplankton taxa, irradiance and nutrient availability
determine the seasonal cycle of DMSP in temperate shelf seas, Mar. Ecol. Prog. Ser., 394, 111–124,
doi:10.3354/meps08284, 2009.
Archer, S. D., Kimmance, S. A., Stephens, J. A., Hopkins, F. E., Bellerby, R. G. J., Schulz, K. G., Piontek, J. and Engel, A.:
Contrasting responses of DMS and DMSP to ocean acidification in Arctic waters, Biogeosciences, 10(3), 1893–1908,
doi:10.5194/bg-10-1893-2013, 2013.
Arnold, H. E., Kerrison, P. and Steinke, M.: Interacting effects of ocean acidification and warming on growth and DMS-
production in the haptophyte coccolithophore Emiliania huxleyi., Glob. Chang. Biol., 19(4), 1007–16,
doi:10.1111/gcb.12105, 2013.
Avgoustidi, V., Nightingale, P. D., Joint, I., Steinke, M., Turner, S. M., Hopkins, F. E. and Liss, P. S.: Decreased marine
dimethyl sulfide production under elevated $CO_2$ levels in mesocosm and in vitro studies, Environ. Chem., 9(4), 399–404,
doi:10.1071/EN11125, 2012.
Barlow, R. G., Cummings, D. G. and Gibb, S. W.: Improved resolution of mono- and divinyl chlorophylls a and b and
zeaxanthin and lutein in phytoplankton extracts using reverse phase C-8 HPLC, Mar. Ecol. Prog. Ser., 161, 303–307, 1997.
Bentley, R. and Chasteen, T. G.: Environmental VOSCs—formation and degradation of dimethyl sulfide, methanethiol and
related materials, Chemosphere, 55(3), 291–317, doi:10.1016/j.chemosphere.2003.12.017, 2004.
Brussaard, C. P. D., Noordeloos, A. A. M., Witte, H., Collenteur, M. C. J., Schulz, K., Ludwig, A. and Riebesell, U.: Arctic
microbial community dynamics influenced by elevated $CO_2$ levels, Biogeosciences, 10(2), 719–731, doi:10.5194/bg-10-
660   719-2013, 2013.

Brutemark, A., Engström-Öst, J. and Vehmaa, A.: Long-term monitoring data reveal pH dynamics, trends and variability in
the western Gulf of Finland, Oceanol. Hydrobiol. Stud., 40(3), 91–94, doi:10.2478/s13545-011-0034-3, 2011.
Butler, A. and Carter-Franklin, J. N.: The role of vanadium bromoperoxidase in the biosynthesis of halogenated marine
natural products, Nat. Prod. Rep., 21(1), 180–188, doi:10.1039/b302337k, 2004.
Canadell, J. G., Le Quéré, C., Raupach, M. R., Field, C. B., Buitenhuis, E. T., Ciais, P., Conway, T. J., Gillett, N. P.,
Houghton, R. A. and Marland, G.: Contributions to accelerating atmospheric $CO_2$ growth from economic activity, carbon
intensity, and efficiency of natural sinks., Proc. Natl. Acad. Sci. U. S. A., 104(47), 18866–18870,
doi:10.1073/pnas.0702737104, 2007.
Carpenter, L. J., Hopkins, J. R., Jones, C. E., Lewis, A. C., Parthipan, R., Wevill, D. J., Poissant, L., Pilote, M. and
Constant, P.: Abiotic source of reactive organic halogens in the sub-arctic atmosphere?, Environ. Sci. Technol., 39(22),
8812–8816, 2005.
Carrión, O., Curson, A. R. J., Kumaresan, D., Fu, Y., Lang, A. S., Mercadé, E. and Todd, J. D.: A novel pathway producing
dimethylsulphide in bacteria is widespread in soil environments, Nat. Commun., 6, 6579, doi:10.1038/ncomms7579, 2015.
Chance, R., Baker, A. R., Küpper, F. C., Hughes, C., Kloareg, B. and Malin, G.: Release and transformations of inorganic
iodine by marine macroalgae, Estuar. Coast. Shelf Sci., 82, 406–414, doi:10.1016/j.ecss.2009.02.004, 2009.
Charlson, R. J., Lovelock, J. E., Andreae, M. O. and Warren, S. G.: Oceanic phytoplankton, atmospheric sulphur, cloud
albedo and climate, Nature, 326(6114), 655–661, 1987.
Chuck, A. L., Turner, S. M. and Liss, P. S.: Oceanic distributions and air-sea fluxes of biogenic halocarbons in the open
ocean, J. Geophys. Res., 110(C10022), doi:10.1029/2004JC002741, 2005.
Collins, M., Knutti, R., Arblaster, J., Dufresne, J.-L., Fichefet, T., Frielingstein, P., Gao, X., Gutowski, W. J., Johns, T.,
Krinner, G., Shongwe, M., Tebaldi, C., Weaver, A. J. and Wehner, M.: Long-term climate change: projections,
commitments and irreversibility, in Climate Change 2013: The Physical Science Basis. Contribution of Working Group 1
to the Fifth Assessment Report of the Intergovernmental Panel on Climate Change, edited by T. . Stocker, D. Qin, G.-K.
Plattner, M. Tignor, S. K. Allen, J. Boschung, A. Nauels, Y. Xia, V. Bex, and P. M. Midgley, Cambridge University Press,




Cambridge, UK., 2013.
Crawfurd, K., Brussaard, C. P. D. and Riebesell, U.: The influence of increasing $CO_2$ on microbial community dynamics in
the Baltic Sea, Biogeosciences, Submitted, 2015.
Cubasch, U., Wuebbles, D., Chen, D., Facchini, M. C., Frame, D., Mahowald, N. and Winther, J.-G.: Introduction, in
Climate Change 2013: The Physical Science Basis. Contribution of Working Group 1 to the Fifth Assessment Report of the
Intergovernmental Panel on Climate Change, edited by T. . Stocker, D. Qin, G.-K. Plattner, M. Tignor, S. K. Allen, J.
Boschung, A. Nauels, Y. Xia, V. Bex, and P. M. Midgley, Cambridge University Press, Cambridge, UK., 2013.
Curran, M. A. J., Jones, G. B. and Burton, H.: Spatial distribution of dimethylsulfide and dimethylsulfoniopropionate in the
Australasian sector of the Southern Ocean, J. Geophys. Res., 103(D13), 16677 – 16689, 1998.
Curson, A. R. J., Todd, J. D., Sullivan, M. J. and Johnston, A. W. B.: Catabolism of dimethylsulphoniopropionate:
microorganisms, enzymes and genes, Nat. Rev. Microbiol., 9(12), 849–859, doi:10.1038/nrmicro2653, 2011.
Czerny, J., Schulz, K. G., Krug, S. A., Ludwig, A. and Riebesell, U.: Technical Note: The determination of enclosed water
volume in large flexible-wall mesocosms "KOSMOS," Biogeosciences, 10, 1937–1941, doi:10.5194/bg-10-1937-2013,
698     2013.

Doney, S. C., Fabry, V. J., Feely, R. A. and Kleypas, J. A.: Ocean acidification: the other $CO_2$ problem., Ann. Rev. Mar.
Sci., 1, 169–192, doi:10.1146/annurev.marine.010908.163834, 2009.
Drotar, A., Burton, G. A., Tavernier, J. E. and Fall, R.: Widespread occurrence of bacterial thiol methyltransferases and the
biogenic emission of methylated sulfur gases, Appl. Environ. Microbiol., 53(7), 1626–1631, 1987.
Elliott, S. and Rowland, F. S.: Methyl halide hydrolysis rates in natural waters, J. Atmos. Chem., 20, 229–236, 1995.
Endres, S., Galgani, L., Riebesell, U., Schulz, K.-G. and Engel, A.: Stimulated bacterial growth under elevated $p$$CO_2$:
results from an off-shore mesocosm study., PLoS One, 9(6), e99228, doi:10.1371/journal.pone.0099228, 2014.
Engel, A., Schulz, K. G., Riebesell, U., Bellerby, R. G. J., Delille, B. and Schartau, M.: Effects of $CO_2$ on particle size
distribution and phytoplankton abundance during a mesocosm bloom experiment (PeECE II), Biogeosciences, 5(2), 509–
521, doi:10.5194/bg-5-509-2008, 2008.
Engel, A., Borchard, C., Piontek, J., Schulz, K. G., Riebesell, U. and Bellerby, R. G. J.: $CO_2$ increases [14]C primary
production in an Arctic plankton community, Biogeosciences, 10(3), 1291–1308, doi:10.5194/bg-10-1291-2013, 2013.
Finni, T., Kononen, K., Olsonen, R. and Wallström, K.: The History of Cyanobacterial Blooms in the Baltic Sea, AMBIO
A J. Hum. Environ., 30(4), 172–178, doi:10.1579/0044-7447-30.4.172, 2001.
Franklin, D. J., Poulton, A. J., Steinke, M., Young, J., Peeken, I. and Malin, G.: Dimethylsulphide, DMSP-lyase activity
and microplankton community structure inside and outside of the Mauritanian upwelling, Prog. Oceanogr., 83(1-4), 134–
142, doi:10.1016/j.pocean.2009.07.011, 2009.
Gidhagen, L.: Coastal upwelling in the Baltic Sea—Satellite and in situ measurements of sea-surface temperatures
indicating coastal upwelling, Estuar. Coast. Shelf Sci., 24, 449–462, 1987.
Goodwin, K., Schaefer, J. K. and Oremland, R. S.: Bacterial oxidation of dibromomethane and methyl bromide in natural
waters and enrichment cultures, Appl. Environ. Microbiol., 64(12), 4629 –4636, 1998.
Goodwin, K. D., North, W. J. and Lidstrom, M. E.: Production of bromoform and dibromomethane by giant kelp: factors
affecting release and comparison to anthropogenic bromine sources, Limnol. Oceanogr., 42(8), 1725–1734,
doi:10.4319/lo.1997.42.8.1725, 1997.
Hall-Spencer, J. M., Rodolfo-Metalpa, R., Martin, S., Ransome, E., Fine, M., Turner, S. M., Rowley, S. J., Tedesco, D. and
Buia, M.-C.: Volcanic carbon dioxide vents show ecosystem effects of ocean acidification, Nature, 454(7200), 96–99,
doi:10.1038/nature07051, 2008.




Hartmann, D. L., Klein Tank, A. M. G., Rusticucci, M., Alexander, L. V., Bronnimann, S., Charabi, Y., Dentener, F. J.,
Dlugokencky, E. J., Easterling, D. R., Kaplan, A., Soden, B. J., Thorne, P. W., Wild, M. and Zhai, P. M.: Observations:
Atmosphere and Surface, in Climate Change 2013: The Physical Science Basis. Contribution of Working Group 1 to the
Fifth Assessment Report of the Intergovernmental Panel on Climate Change, edited by T. F. Stocker, D. Qin, G.-K.
Plattner, M. Tignor, S. K. Allen, J. Boschung, A. Nauels, Y. Xia, V. Bex, and P. M. Midgley, Cambridge University Press,
Cambridge, Cambridge, UK., 2013.
Hatton, A. D., Darroch, L. and Malin, G.: The role of dimethylsulphoxide in the marine biogeochemical cycle of
dimethylsulphide, Oceanogr. Mar. Biol. An Annu. Rev., 42, 29–56, 2004.
Hofmann, G. E., Barry, J. P., Edmunds, P. J., Gates, R. D., Hutchins, D. A., Klinger, T. and Sewell, M. A.: The effect of
ocean acidification on calcifying organisms in marine ecosystems: an organism-to-ecosystem perspective, Annu. Rev. Ecol.
Evol. Syst., 41(1), 127–147, doi:10.1146/annurev.ecolsys.110308.120227, 2010.
Hopkins, F. E. and Archer, S. D.: Consistent increase in dimethyl sulphide (DMS) in response to high $CO_2$ in five
shipboard bioassays from contrasting NW European waters, Biogeosciences, 11(2), 4925 – 4940, doi:10.5194/bgd-11-
739  2267-2014, 2014.

Hopkins, F. E., Turner, S. M., Nightingale, P. D., Steinke, M., Bakker, D. and Liss, P. S.: Ocean acidification and marine
trace gas emissions., Proc. Natl. Acad. Sci. U. S. A., 107(2), 760–765, doi:10.1073/pnas.0907163107, 2010.
Hopkins, F. E., Kimmance, S. A., Stephens, J. A., Bellerby, R. G. J., Brussaard, C. P. D., Czerny, J., Schulz, K. G. and
Archer, S. D.: Response of halocarbons to ocean acidification in the Arctic, Biogeosciences, 10(4), 2331–2345,
doi:10.5194/bg-10-2331-2013, 2013.
Hornick, T., Brussaard, C. P., Crawfurd, K., Spilling, K., Bach, L. T. and Grossart, H.-P.: Testing the effect of $pCO_2$ on
bacterial dynamics during a Phytoplankton bloom in a mesocosm experiment in the Baltic Sea, Biogeosciences, Submitted,
747  2015.

Hughes, C., Malin, G., Nightingale, P. D. and Liss, P. S.: The effect of light stress on the release of volatile iodocarbons by
three species of marine microalgae, Limnol. Oceanogr., 51(6), 2849–2854, 2006.
Hughes, C., Malin, G., Turley, C. M., Keely, B. J., Nightingale, P. D. and Liss, P. S.: The production of volatile
iodocarbons by biogenic marine aggregates, Limnol. Oceanogr., 53(2), 867–872, 2008.
Hughes, C., Chuck, A. L., Rossetti, H., Mann, P. J., Turner, S. M., Clarke, A., Chance, R. and Liss, P. S.: Seasonal cycle of
seawater bromoform and dibromomethane concentrations in a coastal bay on the western Antarctic Peninsula, Global
Biogeochem. Cycles, 23, doi:10.1029/2008GB003268, 2009.
Hughes, C., Franklin, D. J. and Malin, G.: Iodomethane production by two important marine cyanobacteria:
Prochlorococcus marinus (CCMP 2389) and Synechococcus sp. (CCMP 2370), Mar. Chem., 125(1-4), 19–25,
doi:10.1016/j.marchem.2011.01.007, 2011.
Hughes, C., Johnson, M., Utting, R., Turner, S., Malin, G., Clarke, a. and Liss, P. S.: Microbial control of bromocarbon
concentrations in coastal waters of the western Antarctic Peninsula, Mar. Chem., 151, 35–46,
doi:10.1016/j.marchem.2013.01.007, 2013.
Janssen, F., Schrum, C. and Backhaus, J.: A climatological data set of temperature and salinity for the Baltic Sea and the
North Sea, Dtsch. Hydrogr. Zeitschrift, Supplement, 1999.
Kangro, K., Olli, K., Tamminen, T. and Lignell, R.: Species-specific responses of a cyanobacteria-dominated
phytoplankton community to artificial nutrient limitation in the Baltic Sea, Mar. Ecol. Prog. Ser., 336, 15–27, 2007.
Karlsson, A., Auer, N., Schulz-Bull, D. and Abrahamsson, K.: Cyanobacterial blooms in the Baltic — A source of
halocarbons, Mar. Chem., 110, 129–139, doi:10.1016/j.marchem.2008.04.010, 2008.
Kiene, R. P. and Hines, M. E.: Microbial formation of dimethyl sulfide in anoxic sphagnum peat, Appl. Environ.
Microbiol., 61(7), 2720–2726, 1995.





Kiene, R. P. and Visscher, P. T.: Production and fate of methylated sulfur compounds from methionine and
dimethylsulfoniopropionate in anoxic salt marsh sediments., Appl. Environ. Microbiol., 53(10), 2426–2434, 1987.
Kim, J.-M., Lee, K., Yang, E. J., Shin, K., Noh, J. H., Park, K.-T., Hyun, B., Jeong, H.-J., Kim, J.-H., Kim, K. Y., Kim, M.,
Kim, H.-C., Jang, P.-G. and Jang, M.-C.: Enhanced production of oceanic dimethylsulfide resulting from $CO_2$-induced
grazing activity in a high $CO_2$ world., Environ. Sci. Technol., 44(21), 8140–8143, doi:10.1021/es102028k, 2010.
Klick, S.: Seasonal variations of biogenic and anthropogenic halocarbons in seawater from a coastal site, Limnol.
Oceanogr., 37(7), 1579–1585, 1992.
Klick, S. and Abrahamsson, K.: Biogenic volatile iodated hydrocarbons in the ocean, J. Geophys. Res., 97(C8), 12683–
777    12687, 1992.

Lana, A., Simó, R., Vallina, S. M. and Dachs, J.: Re-examination of global emerging patterns of ocean DMS concentration,
Biogeochemistry, 110, 173–182, doi:10.1007/s10533-011-9677-9, 2012.
Laturnus, F., Giese, B., Wiencke, C. and Adams, F. C.: Low-molecular-weight organoiodine and organobromine
compounds released by polar macroalgae-the influence of abiotic factors, Fresenius. J. Anal. Chem., 368, 297–302, 2000.
Leck, C. and Rodhe, H.: Emissions of marine biogenic sulfur to the atmosphere of northern Europe, J. Atmos. Chem., 12,
63–86, 1991.
Leck, C., Larsson, U., Bågander, L. E., Johansson, S. and Hajdu, S.: Dimethyl sulfide in the Baltic Sea: annual variability
in relation to biological activity, J. Geophys. Res., 95(C3), 3353–3363, doi:10.1029/JC095iC03p03353, 1990.
Lee, G., Park, J., Jang, Y., Lee, M., Kim, K. R., Oh, J. R., Kim, D., Yi, H. Il and Kim, T. Y.: Vertical variability of
seawater DMS in the South Pacific Ocean and its implication for atmospheric and surface seawater DMS, Chemosphere,
78(8), 1063–1070, doi:10.1016/j.chemosphere.2009.10.054, 2010.
Leedham, E. C., Hughes, C., Keng, F. S. L., Phang, S.-M., Malin, G. and Sturges, W. T.: Emission of atmospherically
significant halocarbons by naturally occurring and farmed tropical macroalgae, Biogeosciences, 10(6), 3615–3633,
doi:10.5194/bg-10-3615-2013, 2013.
Lehmann, A. and Myrberg, K.: Upwelling in the Baltic Sea — A review, J. Mar. Syst., 74, S3–S12,
doi:10.1016/j.jmarsys.2008.02.010, 2008.
Lischka, S., Riebesell, U., Stuhr, A. and Bermudez, J. R.: Micro- and mesozooplankton community response to increasing
levels of $CO_2$ in the Baltic Sea: insights from a large-scale mesocosm experiment, Biogeosciences, Submitted, 2015.
Liss, P., Marandino, C. A., Dahl, E., Helmig, D., Hintsa, E. J., Hughes, C., Johnson, M., Moore, R. M., Plane, J. M. C.,
Quack, B., Singh, H. B., Stefels, J., von Glasow, R. and Williams, J.: Short-lived trace gases in the surface ocean and the
atmosphere, in Ocean-Atmosphere Interactions of Gases and Particles, edited by P. Liss and M. Johnson, pp. 55–112.,
799    2014.

Lomans, B. P., Smolders, A., Intven, L. M., Pol, A., Op, D. and van der Drift, C.: Formation of dimethyl sulfide and
methanethiol in anoxic freshwater sediments, Appl. Environ. Microbiol., 63(12), 4741–4747, 1997.
Mackey, M. D., Mackey, D. J., Higgins, H. W. and Wright, S. W.: CHEMTAX a program for estimating class abundances
from chemical markers: application to HPLC measurements of phytoplankton, Mar. Ecol. Prog. Ser., 144, 265–283, 1996.
Manley, S. L. and De La Cuesta, J. L.: Methyl iodide production from marine phytoplankton cultures, Limnol. Oceanogr.,
42(1), 142–147, doi:10.4319/lo.1997.42.1.0142, 1997.
Manley, S. L., Goodwin, K. and North, W. J.: Laboratory production of bromoform, methylene bromide, and methyl iodide
by macroalgae in and distribution nearshore Southern California waters, Limnol. Oceanogr., 37(8), 1652–1659, 1992.
Martino, M., Liss, P. S. and Plane, J. M. C.: The photolysis of dihalomethanes in surface seawater., Environ. Sci. Technol.,
39(18), 7097–7101, 2005.



Moore, R. M.: Methyl halide production and loss rates in sea water from field incubation experiments, Mar. Chem., 101(3-
4), 213–219, doi:10.1016/j.marchem.2006.03.003, 2006.
Moore, R. M. and Tokarczyk, R.: Volatile biogenic halocarbons in the northwest Atlantic, Global Biogeochem. Cycles,
7(1), 195–210, 1993.
Moore, R. M. and Zafiriou, O. C.: Photochemical production of methyl iodide in seawater, J. Geophys. Res., 99(D8)
16415–16420, doi:10.1029/94JD00786, 1994.
National Measurement Institute of Australia: Proficiency Study 12-23: DMS in seawater., 2013.
Nausch, M., Nausch, G., Lass, H. U., Mohrholz, V., Nagel, K., Siegel, H. and Wasmund, N.: Phosphorus input by
upwelling in the eastern Gotland Basin (Baltic Sea) in summer and its effects on filamentous cyanobacteria, Estuar. Coast.
Shelf Sci., 83(4), 434–442, doi:10.1016/j.ecss.2009.04.031, 2009.
Newbold, L. K., Oliver, A. E., Booth, T., Tiwari, B., DeSantis, T., Maguire, M., Andersen, G., van der Gast, C. J. and
Whiteley, A. S.: The response of marine picoplankton to ocean acidification, Environ. Microbiol., 14(9), 2293–2307,
doi:10.1111/j.1462-2920.2012.02762.x, 2012.
Niemisto, L., Rinne, I. and Melvasalo, T.: Blue-green algae and their nitrogen fixation in the Baltic Sea in 1980, 1982 and
1984, Meri, 17, 1–59, 1989.
O'Dowd, C. D., Jimenez, J. L., Bahreini, R., Flagan, R. C., Seinfeld, J. H., Hameri, K., Pirjola, L., Kulmala, M., Jennings,
S. G. and Hoffmann, T.: Marine aerosol formation from biogenic iodine emissions, Nature, 417(6889), 632–636,
doi:10.1038/nature00773.1.2.3.4.5.6.7.8.9.10., 2002.
Oberg, J.: Cyanobacterial blooms in the Baltic Sea in 2013, HELCOM Balt. Sea Environ. Fact Sheet, 2013.
Orlikowska, A. and Schulz-Bull, D. E.: Seasonal variations of volatile organic compounds in the coastal Baltic Sea,
Environ. Chem., 6, 495–507, doi:10.1071/EN09107, 2009.
Park, K.-T., Lee, K., Shin, K., Yang, E. J., Hyun, B., Kim, J.-M., Noh, J. H., Kim, M., Kong, B., Choi, D. H., Choi, S.-J.,
Jang, P.-G. and Jeong, H. J.: Direct linkage between dimethyl sulfide production and microzooplankton grazing, resulting
from prey composition change under high partial pressure of carbon dioxide conditions., Environ. Sci. Technol., 48(9),
4750–4756, doi:10.1021/es403351h, 2014.
Passow, U. and Riebesell, U.: Mesocosm perturbation experiments and the sensitivity of marine biological systems to
global change, Solas News, (1), 12–13, doi:10.1029/2003JC002120, 2005.
Paul, A. J., Bach, L. T., Schulz, K.-G., Boxhammer, T., Czerny, J., Achterberg, E., Hellemann, D., Trense, Y., Nausch, M.,
Sswat, M. and Riebesell, U.: Effect of elevated $CO_2$ on organic matter pools and fluxes in a summer, post spring-bloom
Baltic Sea plankton community., Biogeosciences, 12, 6181 – 6203, 2015.
Pedersen, M., Collen, J., Abrahamsson, K. and Ekdahl, A.: Production of halocarbons from seaweeds: an oxidative stress
reaction?, Sci. Mar., 60(Supplement 1), 257–263, 1996.
Piontek, J., Borchard, C., Sperling, M., Schulz, K. G., Riebesell, U. and Engel, A.: Response of bacterioplankton activity in
an Arctic fjord system to elevated $pCO_2$: results from a mesocosm perturbation study, Biogeosciences, 10, 297–314,
doi:10.5194/bg-10-297-2013, 2013.
Quack, B., Peeken, I., Petrick, G. and Nachtigall, K.: Oceanic distribution and sources of bromoform and dibromomethane
in the Mauritanian upwelling, J. Geophys. Res., 112, C10006, doi:10.1029/2006JC003803, 2007.
Quinn, P. K. and Bates, T. S.: The case against climate regulation via oceanic phytoplankton sulphur emissions., Nature,
480(7375), 51–56, doi:10.1038/nature10580, 2011.
Raateoja, M., Kuosa, H. and Hällfors, S.: Fate of excess phosphorus in the Baltic Sea: A real driving force for
cyanobacterial blooms?, J. Sea Res., 65(2), 315–321, doi:10.1016/j.seares.2011.01.004, 2011.





Raven, J. R., Caldeira, K., Elderfield, H., Hoegh-Guldberg, O., Liss, P. S., Riebesell, U., Shepherd, J., Turley, C. and
Watson, A.: Ocean acidification due to increasing atmospheric carbon dioxide, R. Soc. Policy Doc. 12/05, (June), 2005.
Riebesell, U., Czerny, J., von Bröckel, K., Boxhammer, T., Büdenbender, J., Deckelnick, M., Fischer, M., Hoffmann, D.,
Krug, S. A., Lentz, U., Ludwig, A., Muche, R. and Schulz, K. G.: Technical Note: A mobile sea-going mesocosm system –
new opportunities for ocean change research, Biogeosciences, 10(3), 1835–1847, doi:10.5194/bg-10-1835-2013, 2013.
Ross, P. M., Parker, L., O'Connor, W. A. and Bailey, E. A.: The impact of ocean acidification on reproduction, early
development and settlement of marine organisms, Water, 3(4), 1005–1030, doi:10.3390/w3041005, 2011.
Roy, R., Pratihary, A., Narvenkar, G., Mochemadkar, S., Gauns, M. and Naqvi, S. W. A.: The relationship between volatile
halocarbons and phytoplankton pigments during a Trichodesmium bloom in the coastal eastern Arabian Sea, Estuar. Coast.
Shelf Sci., 95(1), 110–118, doi:10.1016/j.ecss.2011.08.025, 2011.
Scarratt, M. G. and Moore, R. M.: Production of methyl bromide and methyl chloride in laboratory cultures of marine
phytoplankton II, Mar. Chem., 59(3-4), 311–320, doi:10.1016/S0304-4203(97)00092-3, 1998.
Schall, C., Heumann, K. G. and Kirst, G. O.: Biogenic volatile organoiodine and organobromine hydrocarbons in the
Atlantic Ocean from 42°N to 72°S, Fresenius. J. Anal. Chem., 359(3), 298–305, doi:10.1007/s002160050577, 1997.
Schneider, B., Nausch, G., Kubsch, H. and Petersohn, I.: Accumulation of total $CO_2$ during stagnation in the Baltic Sea
deep water and its relationship to nutrient and oxygen concentrations, Mar. Chem., 77, 277–291, 2002.
Schulz, K. G., Bellerby, R. G. J., Brussaard, C. P. D., Büdenbender, J., Czerny, J., Engel, A., Fischer, M., Koch-Klavsen,
S., Krug, S. A., Lischka, S., Ludwig, A., Meyerhöfer, M., Nondal, G., Silyakova, A., Stuhr, A. and Riebesell, U.: Temporal
biomass dynamics of an Arctic plankton bloom in response to increasing levels of atmospheric carbon dioxide,
Biogeosciences, 10(1), 161–180, doi:10.5194/bg-10-161-2013, 2013.
Simó, R., Archer, S. D., Pedros-Alio, C., Gilpin, L. and Stelfox-Widdicombe, C. E.: Coupled dynamics of
dimethylsulfoniopropionate and dimethylsulfide cycling and the microbial food web in surface waters of the North
Atlantic, Limnol. Oceanogr., 47(1), 53–61, 2002.
Simó, R., Vila-Costa, M., Alonso-Sáez, L., Cardelús, C., Guadayol, Ó., Vázquez-Dominguez, E. and Gasol, J. M.: Annual
DMSP contribution to S and C fluxes through phytoplankton and bacterioplankton in a NW Mediterranean coastal site,
Aquat. Microb. Ecol., 57(October), 43–55, doi:10.3354/ame01325, 2009.
Six, K. D., Kloster, S., Ilyina, T., Archer, S. D., Zhang, K. and Maier-Reimer, E.: Global warming amplified by reduced
sulphur fluxes as a result of ocean acidification, Nat. Clim. Chang., 3(8), 1–4, doi:10.1038/nclimate1981, 2013.
Smith, D. C., Simon, M., Alldredge, A. L. and Azam, F.: Intense hydrolytic enzyme activity on marine aggregates and
implications for rapid particle dissolution, Nature, 359, 139 – 142, 1992.
Solomon, S., Garcia, R. R. and Ravishankara, A. R.: On the role of iodine in ozone depletion, J. Geophys. Res., 99(D10),
20491–20499, doi:10.1029/94JD02028, 1994.
Stal, L. J., Albertano, P., Bergman, B., von Bröckel, K., Gallon, J. R., Hayes, P. K., Sivonen, K. and Walsby, A. E.:
BASIC: Baltic Sea cyanobacteria. An investigation of the structure and dynamics of water blooms of cyanobacteria in the
Baltic Sea—responses to a changing environment, Cont. Shelf Res., 23(17-19), 1695–1714, doi:10.1016/j.csr.2003.06.001,
886  2003.

Stets, E. G., Hines, M. E. and Kiene, R. P.: Thiol methylation potential in anoxic, low-pH wetland sediments and its
relationship with dimethylsulfide production and organic carbon cycling., FEMS Microbiol. Ecol., 47(1), 1–11,
doi:10.1016/S0168-6496(03)00219-8, 2004.
Sunda, W., Kieber, D. J., Kiene, R. P. and Huntsman, S.: An antioxidant function for DMSP and DMS in marine algae,
Nature, 418(6895), 317–320, doi:10.1038/nature00851, 2002.
Theiler, R., Cook, J. C., Hager, L. P. and Siuda, J. F.: Halohydrocarbon synthesis by bromoperoxidase, Science.,




202(December), 1094 – 1096, 1978.
Turner, S. M., Malin, G., Liss, P. S., Harbour, D. S. and Holligan, P. M.: The seasonal variation of dimethyl sulfide and
dimethylsulfoniopropionate concentrations in nearshore waters, Limnol. Oceanogr., 33(3), 364–375, 1988.
Urhahn, T. and Ballschmiter, K.: Chemistry of the biosynthesis of halogenated methanes: C1-organohalogens as pre-
industrial chemical stressors in the environment?, Chemosphere, 37(6), 1017–1032, doi:10.1016/S0045-6535(98)00100-3,
898  1998.

del Valle, D. A., Slezak, D., Smith, C. M., Rellinger, A. N., Kieber, D. J. and Kiene, R. P.: Effect of acidification on
preservation of DMSP in seawater and phytoplankton cultures: Evidence for rapid loss and cleavage of DMSP in samples
containing Phaeocystis sp., Mar. Chem., 124, 57–67, doi:10.1016/j.marchem.2010.12.002, 2011.
Vila-Costa, M., Simó, R., Harada, H., Gasol, J. M., Slezak, D. and Kiene, R. P.: Dimethylsulfoniopropionate uptake by
marine phytoplankton, Science, 314(5799), 652–4, doi:10.1126/science.1131043, 2006a.
Vila-Costa, M., del Valle, D. A., González, J. M., Slezak, D., Kiene, R. P., Sánchez, O. and Simó, R.: Phylogenetic
identification and metabolism of marine dimethylsulfide-consuming bacteria., Environ. Microbiol., 8(12), 2189–2200,
doi:10.1111/j.1462-2920.2006.01102.x, 2006b.
Visscher, P. T., Baumgartner, L. . K., Buckley, D. H., Rogers, D. R., Hogan, M. E., Raleigh, C. D., Turk, K. A. and Des
Marais, D. J.: Dimethyl sulphide and methanethiol formation in microbial mats: potential pathways for biogenic signatures,
Environ. Microbiol., 5(4), 296–308, 2003.
Vogt, M., Steinke, M., Turner, S. M., Paulino, A., Meyerhöfer, M., Riebesell, U., LeQuéré, C. and Liss, P. S.: Dynamics of
dimethylsulphoniopropionate and dimethylsulphide under different $CO_2$ concentrations during a mesocosm experiment,
Biogeosciences, 5(2), 407–419, doi:10.5194/bg-5-407-2008, 2008.
Wasmund, N.: Occurrence of cyanobacterial blooms in the Baltic Sea in relation to environmental conditions, Iny. Rev.
ges. Hydrobiol., 82(2), 169–184, 1997.
Webb, A.: The effects of elevated $CO_2$ and ocean acidification on the production of marine biogenic trace gases, PhD
Thesis, Univ. East Angl., (March), 2015.
Webb, A. L., Malin, G., Hopkins, F. E., Ho, K.-L., Riebesell, U., Schulz, K., Larsen, A. and Liss, P.: Ocean acidification
has different effects on the production of dimethylsulphide and dimethylsulphoniopropionate measured in cultures of
Emiliania huxleyi RCC1229 anda mesocosm study: a comparison of laboratory monocultures and community interactions,
Environ. Chem., EN14268, doi:http://dx.doi.org/10.1071/EN14268, 2015.
Welschmeyer, N. A.: Fluorometric analysis of chlorophyll a in the presence of chlorophyll b and pheopigments, Limnol.
Oceanogr., 39(8), 1985–1992, 1994.
Zika, R. G., Gidel, L. T. and Davis, D. D.: A comparison of photolysis and substitution decomposition rates of methyl
iodide in the ocean, Geophys. Res. Lett., 11(4), 353–356, 1984.
Zinder, S. H., Doemel, W. N. and Brock, T. D.: Production of volatile sulfur compounds during the decomposition of algal
mats, Appl. Environ. Microbiol., 34(6), 859–861, 1977.
Zindler, C., Peeken, I., Marandino, C. A. and Bange, H. W.: Environmental control on the variability of DMS and DMSP in
the Mauritanian upwelling region, Biogeosciences, 9, 1041–1051, doi:10.5194/bg-9-1041-2012, 2012.






Table 1. Summary of $f$CO$_2$ and pH$_T$ (total scale) during phases 0, 1 and 2 of the mesocosm experiment.

| Mesocosm[a] | Target $f$CO$_2$ / µatm | Whole Experiment / $t$-3 to $t$31 | | Phase 0 / $t$-3 to $t$0 | | Phase I / $t$1 –$t$16 | | Phase II / $t$16 – $t$31 | |
| | | Mean $f$CO$_2$ / µatm | Mean pH / pH$_T$ | Mean $f$CO$_2$ / µatm | Mean pH / pH$_T$ | Mean $f$CO$_2$ / µatm | Mean pH / pH$_T$ | Mean $f$CO$_2$ / µatm | Mean pH / pH$_T$ |
|---|---|---|---|---|---|---|---|---|---|
| M1 | Control | 331 | 7.91 | 231 | 8.00 | 328 | 7.95 | 399 | 7.86 |
| M5 | Control | 334 | 7.91 | 244 | 7.98 | 329 | 7.94 | 399 | 7.52 |
| M7 | 390 | 458 | 7.80 | 239 | 7.99 | 494 | 7.81 | 532 | 7.76 |
| M6 | 840 | 773 | 7.63 | 236 | 7.99 | 932 | 7.59 | 855 | 7.59 |
| M3 | 1120 | 950 | 7.56 | 243 | 7.98 | 1176 | 7.51 | 1027 | 7.52 |
| M8 | 1400 | 1166 | 7.49 | 232 | 8.00 | 1481 | 7.43 | 1243 | 7.45 |
| Baltic Sea | 380 | 350 | 7.91 | 298 | 7.91 | 277 | 7.98 | 436 | 7.86 |

[a] listed in order of increasing $f$CO$_2$





Table 2. Calibration ranges and calculated percentage mean relative standard error for the trace gases
measured in the mesocosms.

| Compound | Calibration range / pmol L$^{-1}$ | % Mean relative standard error |
|---|---|---|
| DMS | 600 – 29300* | 6.33 |
| DMSP | 2030 – 405900* | |
| CH$_3$I | 0.11 – 11.2 | 4.62 |
| CH$_2$I$_2$ | 5.61 – 561.0 | 4.98 |
| C$_2$H$_5$I | 0.10 – 4.91 | 5.61 |
| CH$_2$ClI | 1.98 – 99.0 | 3.64 |
| CHBr$_3$ | 8.61 – 816.0 | 4.03 |
| CH$_2$Br$_2$ | 0.21 – 20.9 | 5.30 |
| CHBr$_2$Cl | 0.07 – 7.00 | 7.20 |

* throughout the rest of this paper, these measurements are given in nmol L$^{-1}$.



Table 3. Abundance and contributions of different phytoplankton groups to the total phytoplankton
community assemblage, showing the range of measurements from total Chl-*a* (Paul *et al.,* 2015),
CHEMTAX analysis of derived Chl-*a* (Paul *et al.*, 2015) and phytoplankton abundance (Crawfurd *et*
*al.*, 2015). Data are split into the range of all the mesocosm measurements and those from the Baltic
Sea.

| | Mesocosm | | | Baltic Sea | | |
|---|---|---|---|---|---|---|
| | Range Integrated 10 m | Range Integrated 17 m | % Contribution to Chl-*a* | Range Integrated 10 m | Range Integrated 17 m | % Contribution to Chl-*a* |
| **Chl-*a*** | 0.9 – 2.9 | 0.9 – 2.6 | 100 | 1.3 – 6.5 | 1.12 – 5.5 | 100 |
| **Phytoplankton Taxonomy / Equivalent Chlorophyll µg L$^{-1}$** | | | | | | |
| **Cyanobacteria** | | 0.01 – 0.4 | 8 | | 0.0 – 0.1 | 1 |
| **Prasinophytes** | | 0.04 – 0.3 | 7 | | 0.01 – 0.3 | 4 |
| **Euglenophytes** | | 0.0 – 1.6 | 15 | | 0.0 – 2.6 | 21 |
| **Dinoflagellates** | | 0.0 – 0.3 | 3 | | 0.04 – 0.6 | 9 |
| **Diatoms** | | 0.1 – 0.3 | 7 | | 0.04 – 0.9 | 9 |
| **Chlorophytes** | | 0.3 – 2.0 | 40 | | 0.28 – 3.1 | 41 |
| **Cryptophytes** | | 0.1 – 1.4 | 21 | | 0.1 – 1.0 | 15 |
| **Small Phytoplankton (<10 µm) abundance / cells mL$^{-1}$** | | | | | | |
| **Cyanobacteria** | 55000 – 380000 | 65000 – 470000 | | 30000 – 180000 | 30000 – 250000 | |
| **Picoeukaryotes I** | 15000 – 100000 | 17000 – 111000 | | 5000 – 70000 | 6100 – 78000 | |
| **Picoeukaryotes II** | 700 – 4000 | 600 – 4000 | | 400 – 3000 | 460 – 3700 | |
| **Picoeukaryotes III** | 1000 - 9000 | 1100 – 8500 | | 1000 – 6000 | 950 – 7500 | |
| **Nanoeukaryotes I** | 400 – 1400 | 270 – 1500 | | 200 – 4000 | 210 – 4100 | |
| **Nanoeukaryotes II** | 0 – 400 | 4 – 400 | | 100 – 1100 | 60 – 1300 | |




Table 4. Concentration ranges of trace gases measured in the mesocosms compared to other open
water ocean acidification experiments, showing the range of concentrations for each gas and the
percentage change between the control and the highest $f$CO$_2$ treatment.

| | Range $f$CO$_2$ | | DMS | CH$_3$I | CH$_2$I$_2$ | CH$_2$ClI | CHBr$_3$ | CH$_2$Br$_2$ | CH$_2$Br$_2$Cl |
|---|---|---|---|---|---|---|---|---|---|
| | / µatm | | / nmol L$^{-1}$ | / pmol L$^{-1}$ | | | | | |
| SOPRAN Tvärminne Mesocosm (this study) | 346 – 1333 | Range | 2.7-6.8 | 2.9-6.4 | 57-202 | 3.8-8.0 | 69-148 | 4.0-7.7 | 1.7-3.1 |
| | | % change | -34 | -0.3 | 1.3 | -11 | -9 | -3 | -4 |
| SOPRAN Bergen 2011 (Webb *et al.*, 2015) | 280 – 3000 | Range | 0.1-4.9 | 4.9-32 | 5.8-321 | 9.0-123 | 64-306 | 6.3-30.8 | 3.9-14 |
| | | % change | -60 | -37 | -48 | -27 | -2 | -4 | -6 |
| NERC Microbial Metagenomics Experiment, Bergen 2006 (Hopkins *et al.*, 2010) | 300 - 750 | Range | ND-50 | 2.0-25 | ND-750 | ND-700 | 5.0-80 | ND-5.5 | 0.2-1.2 |
| | | % change | -57 | -41 | -33 | -28 | 13 | 8 | 22 |
| EPOCA Svalbard 2010 (Archer *et al.*, 2013; Hopkins *et al.*, 2013) | 180 - 1420 | Range | ND-14 | 0.04-10 | 0.01-2.5 | 0.3-1.6 | 35-151 | 6.3-33.3 | 1.6-4.7 |
| | | % change | -60 | NS | | NS | NS | NS | NS |
| UKOA European Shelf 2011 (Hopkins and Archer, 2014) | 340 - 1000 | Range | 0.5-12 | | | | | | |
| | | % change | 225 | | | | | | |
| Korean Mesocosm Experiment 2012 (Park *et al.*, 2014) | 160 - 830 | Range | 1.0-100 | | | | | | |
| | | % change | -82 | | | | | | |





Table 5. Concentration ranges of trace gases measured in the Baltic Sea compared to concentrations
measured in the literature. ND – Not Detected.

| Study | DMS concentration range / nmol L$^{-1}$ | Halocarbon concentration range / pmol L$^{-1}$ | | | | | | |
|---|---|---|---|---|---|---|---|---|
| | | CH$_3$I | CH$_2$I$_2$ | C$_2$H$_5$I | CH$_3$ClI | CHBr$_3$ | CH$_2$Br$_2$ | CH$_2$Br$_2$Cl |
| **SOPRAN Tvärminne Baltic Sea (This Study)** | 1.9-11 | 4.3-8.6 | 66.9-374 | 0.6 – 1.0 | 7.0-18 | 93-192 | 7.1-10 | 3.3-5.0 |
| **Orlikowska and Schulz-Bull5(2009)** | 0.3-120 | 1-16 | 0-85 | 0.4 – 1.2 | 5-50 | 5.0-40 | 2.0-10 | 0.8-2.5 |
| **Karlsson *et al.* (2008)** | | 3.0-7.5 | | | | 35-60 | 4.0-7.0 | 2.0-6.5 |
| **Klick and Abrahamsson (1992)** | | | 15-709 | | 11-74 | 14-585 | | |
| **Klick (1992)** | | | ND-243 | | ND-57 | 40-790 | ND-86 | ND-29 |
| **Leck and Rodhe (1991)** | 0.4-2.8 | | | | | | | |
| **Leck *et al.* (1990)** | ND-3.2 | | | | | | | |






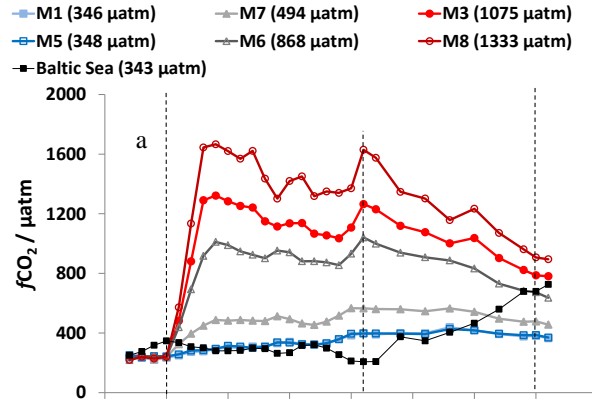


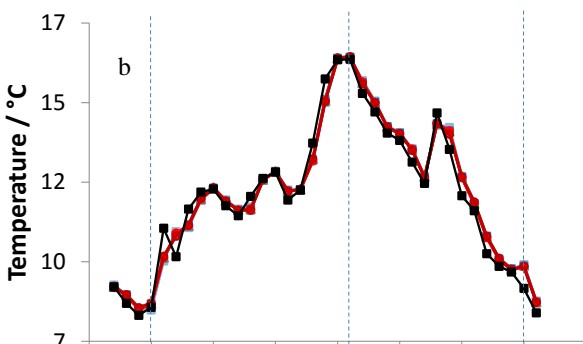


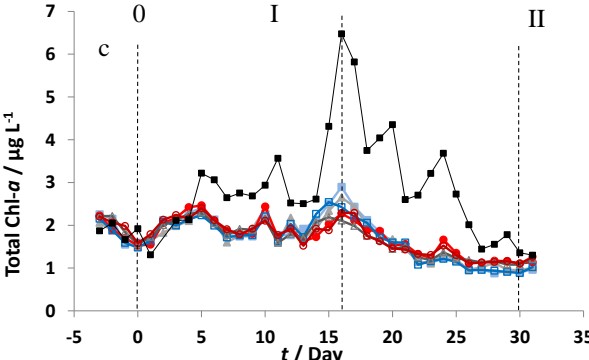

Figure 1. Daily measurements of (a) $f$CO$_2$, (b) mean temperature and (c) total Chlorophyll-a in the
mesocosms and surrounding Baltic Sea waters. Dashed lines represent the three Phases of the
experiment, based on the Chl-$a$ data.




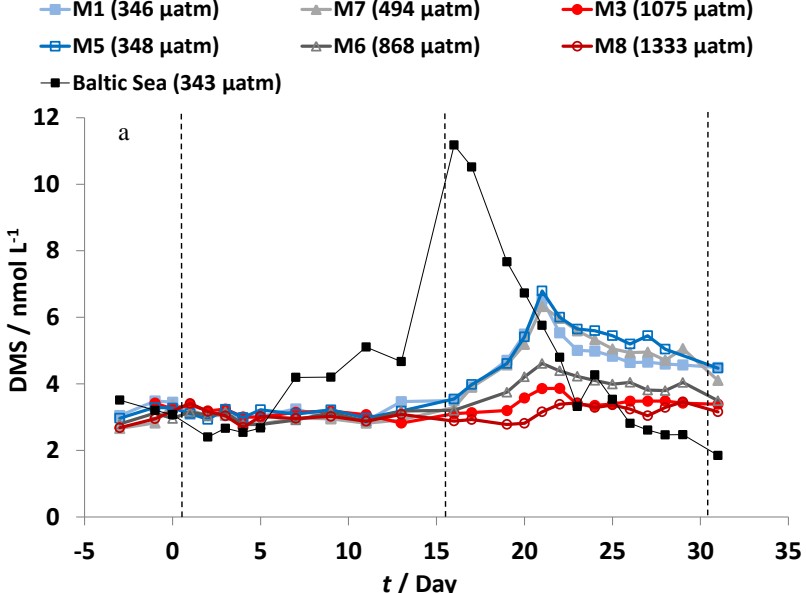


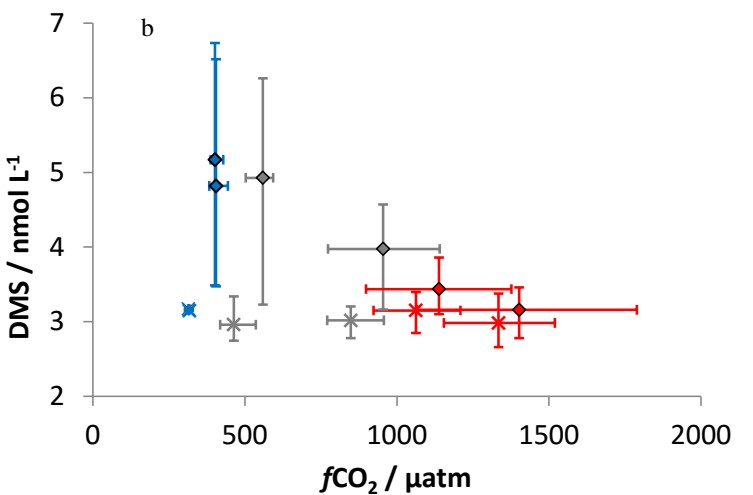


Figure 3. (a) Integrated DMS concentrations measured daily in the mesocosms and Baltic Sea from the
surface 10 m and (b) mean DMS concentrations from each mesocosm during Phase I (crosses) and
Phase II (diamonds), for ambient (blue), medium (grey) and high $f$CO$_2$ (red), with error bars showing
the range of both the DMS and $f$CO$_2$. Dashed lines show the Phases of the experiment as given in Fig.
2, $f$CO$_2$ shown in the legend are mean $f$CO$_2$ across the duration of the experiment.



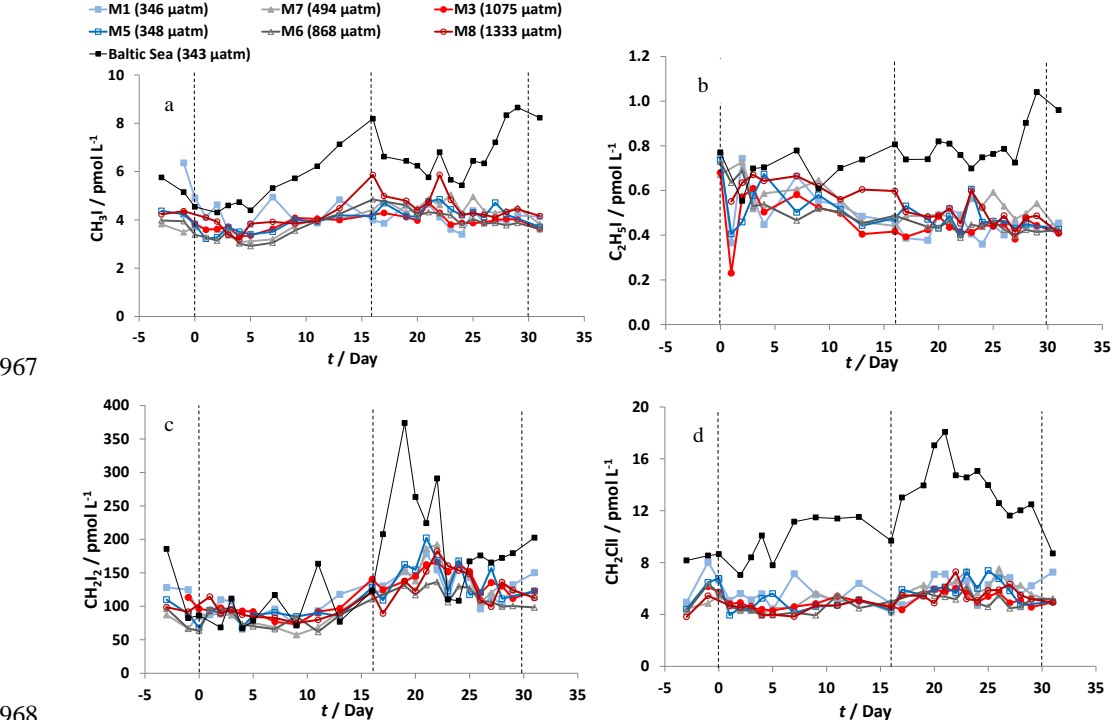



Figure 4. Concentrations (pmol L$^{-1}$) of (a) CH$_3$I, (b) C$_2$H$_5$I, (c) CH$_2$I$_2$ and (d) CH$_2$ClI. Dashed lines indicate the Phases of the experiment, as given in Fig. 2. $f$CO$_2$ shown in the legend are mean $f$CO$_2$ across the duration of the experiment.




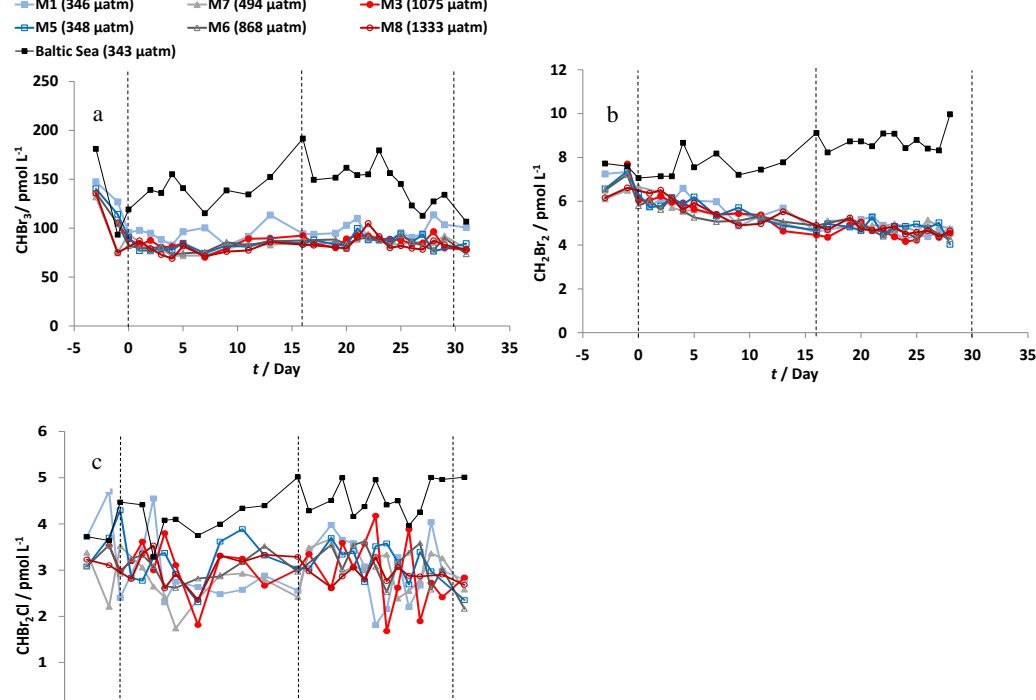



Figure 5. Concentrations (pmol L$^{-1}$) of (a) CHBr$_3$, (b) CH$_2$Br$_2$ and (c) CHBr$_2$Cl. Dashed lines indicate
the phases of the experiment as defined in Fig. 2, $f$CO$_2$ shown in the legend are mean $f$CO$_2$ across the
duration of the experiment.