# Peer review of "Effect of ocean acidification and elevated fCO2 on trace gas"

_Biogeosciences, 2015_

## Referee Comment (RC1) · Anonymous Referee #1 · 7 Mar 2016

Review of "Effect of ocean acidification and elevated fCO2 on trace gas production by a Baltic Sea summer phytoplankton community" Author(s): A.L. Webb et al. MS No.: bg-2015-573

**1-An initial paragraph or section evaluating the overall quality of the discussion paper ("general comments"),**

The manuscript is well structured and for the most parts easily readable. The results show a lack of response of gas concentrations to the experimental design and no linkage to the external conditions due to the outside undergoingits own "experiment", i.e., upwelling.

No rates are reported; not clear any were measured. Hence, the entire manuscript must be clarified that the values represent "net" values and not production, nor consumption or degradation, nor emission.

Hence, I strongly suggest removing most comments from the discussion that pertain to "climate change" and simply state that concentrations remained the same regardless and emphasize that we (and especially modelers) need to have rates of production, consumption, (photo/chem)-degradation or even "net" rates to include in our prognostic and predictive models.

I recommend publishing pending changes. I also suggest shortening some of the longish speculative paragraphs; it's hard to explain why there is no apparent change! h In general, I think the manuscript would profit if it was a bit more structured based on hypotheses, rather than being purely descriptive. You must have had some expectations when the experiment was started (and the proposal written!), especially since you had results from previous mesocosm experiments. Especially since no (?) rates were measured.

**2- section addressing individual scientific questions/issues ("specific comments"),**

The manuscript addresses the influence of ocean acidification on the production of dimethylsulfide (DMS) and 7 halocarbons in a Baltic Sea mesocosm experiment.

The authors effectively found no differences in DMS and halocarbon concentrations over time among the various fCO2 treatments; and no obvious relationship to any other environmental (biological or chemical) variable measured. Difficult to explain without knowing whether turnover is fast. The authors found a decrease of DMS concentrations for highest  $fCO_2$  treatments vs. controls only in the last phase (when Chla declined) and none of the other detected differences in halocarbons were  $CO_2$  related.

The outcome of this study is a relevant piece of information, indicating that most likely there will be no major changes to halocarbon concentrations in the Baltic Sea anytime soon, and the authors conclude that this might be due to the already well adapted community in the unstable Baltic Sea environment with regards to S, T,  $CO_2$  and many other factors. The results are interesting by themselves, and valuable for modelers, though modelers need rates.

The DMS results again confirm results from a range of mesocosm studies.

3- ompact listing of purely technical corrections at the very end ("technical corrections": typing errors, etc.).

Line 247, 248: Inconsistent placing of units, 10m (no space) but 486 nm (space), e.g. line 247, 248 vs 261. You might wanna check if there are more.

Line 70: "Both DMS and DMSP are major routes of sulphur and carbon flux through the marine microbial food web". I wouldn't call them a route, they could be called transporters, or they provide the basis for major routes. Or DMS and DMSP based metabolic pathways are the route...

Line 72: Where do they state that in this reference? I think Simo *et al.*, 2009 should be the reference for phytoplanktonic demand (pages 50-51 e.g.), and Vila Costa *et al.*, 2006a for bacteria (page 653)? Or you put them in as combined references for both (after sulphur demand)

Line 142: That was the standard deviation at the beginning?  $\sim$ 50%,  $\sim$ 7%, and  $\sim$ 75%? That's a lot to start with... Any thoughts on how that could potentially affect the outcome of the experiments on the bacterial metabolism side of halocarbon and DMS production?

Line 169: replace ; with a,

Line 171: why no comparable pigment analysis, what's the rationale behind it?

Line 179: Is that shown anywhere? Otherwise please state what the precision was, and that it is not further shown.

Line 237: careful. The del Valle et al samples were DMSPd and DMSPt; not DMSPp. The Kiene group estimates DMSPp by difference between the total and dissolved pool.

Line 247: 17mIWS space needed Line 248: Chl-a the a is superscripted

Line 300: Mixing of the mesocosms after closure prior to t-3 did not trigger a notable increase in Chl-a in Phase 301 0; in previous mesocosm experiments, mixing redistributed nutrients from the deeper stratified layers 302 throughout the water column

I get what you are saying, but I think you should add what redistributing nutrients did- I am assuming here that it lead to an increase in Chl-a?

Line 282: "mainly through air-sea gas exchange" – isn't that usually considered to be limited by the small surface area / volume ratio? Please comment on why this should not be the same for your analyzed gases.

Line 302: no direct result of the  $CO_2$  additions because there was no significant difference between controls and treatments?

Line 309: chlorophytes (largest contributor to chl a) are not exactly known to be high DMSP or DMS producers; you may want to mention that given stated link to pico and nanoeukaryotes as possible sources. This is why bringing in the Fig, S3 as Fig 3c somewhere actually shows that there are differences among treatments.

Line 311-312: so between the opposing trends for pico I and pico II, the next effect on DMS in the system is zero?

Line 331: Please explain F-test or at least the H0 you used in one sentence in the methods section.

Line 348-369: Simply there was no relationship between patterns (or lack thereof) in DMS concentrations and any other measured variable. And no rate measurements available. Please say so. Too many possibilities, too many unknowns. This section reads a bit like "filler"; sorry.

Line 354: synthesis should be synthesise

Line 358: Correlations between DMS and the cyanobacterial equivalent Chl-a 359 (p=0.42, p<0.01) indicate that the methylation pathway may be a potential source of DMS within the 360 Baltic Sea community.

Reference? Data shown anywhere?

Line 367: Stop! What rates of net DMS production? Did you measured or estimate them? If you did, please indicate and discuss!

Line 371: but I thought that Syn does not make DMS?! There never is high DMS concentration reported along with it in subtrop regions (DiTullio et al., others). Didn't Vla-Costa et al. 2006 report uptake of DMSPd (not DMS) by Syn and other picoeukaryotes?

Line 372: Why is it unlikely? Line 379: just one period.

Line 386: "However, these experiments limit our ability to generalize"... I don't think it's the experiments limiting, but rather the varying responses, is that what you are saying?

Line 410-411: no data on consumption, no bacterial rates described, then what is the basis for this statement? Confusing.

Line 412: "Synechococcus has been identified 412 as a DMS consumer in the open ocean" Reference, please. Syn consumed labelled DMSPd, not DMS (Vila-Costa et al 2006)

Line 431: Sections 3.3 and 3.4: No rates of anything for the halogenated compounds either? Just checking.

Lines 518-522: well, was the region isolated from the coastal environment or not? You can't have it both ways. I understand that the mesocosm bags were closed so they wouldn't have a macroalgal component. This will come back in the discussion

Line 548: I agree that the comparison between the mesocosms and the outside is inappropriate. The outside underwent its own and different "experiment"

Line 557: please delete sentence about DMSp as it implies that there was none because none detected when it is an analytical issue

Line 558-569: given the statement in Line 548, please remove this paragraph as it mixes mesocosm conditions with outside conditions. It is pure speculation as a lot more changed with the injection of upwelled water than fCO2- i.e., particles, nutrients, DOM, etc, etc

Line 576: Is CH2ClI really polyiodinated?

Line 584: Check your manuscript for Chl-a, the a is alternating between superscript and normal

Line 586-590: It is above indicated that macroalgal beds were not a source. Now, it is implied that those macroalgals beds were close? or far? in location w/r to the mesocosms. And the prevailing circulation was from the beds towards the mesocosms?

And waht about vertical input?

The entire DMS section is predicated on upwelling, ie, water injection from below NOT lateral advection. Can't have tvertical input for one gas and horizontal input for the other one.

Line 593-607: good

Line 599: I think you want to stress here, that the values are high enough to be considered an already adapted site, rather than stressing that they are lower than elsewhere, correct? "[...] at such a location with a relatively low fCO2 excursion compared to some sites [...]", maybe rephrase to "[...] at such a location with a relatively high fCO2 excursion, however still relatively low when compared to some sites [...]"

Line 609-611: Not all the time, only after the decline of Chl-a, right? I wouldn't stretch it out, then.

Line 614: production was not measured, only concentrations. Please change production for cycling because the levels measured are a net result

Line 615: since rates were not measured, you don't know whether was a response (ie, prodn and/or cons), only that the measured concentrations did not change

Line 617: no change IF under similar meteorological conditions as during this sampling

Line 617-621: NET production or availability. Again, same issue. Also, rather simplistic as meteorology must be considered.

L621-625: This is a weak concluding paragraph. It says nothing at all. Keep it honest and simple by saying that no changes in concetrations were seen and that next time it would be best to measure rates so these rates can be included in models to have better predictions!! So sorry that you didn't see any changes nor anything "exciting".

Figures in general:

I find it very irritating how the units are given, e.g. fCO2/µatm. I read the "/" as 'per', which makes it confusing. I would very much prefer if you put fCO2 (µatm) or fCO2 [µatm]

Fig 3: The Legend is misleading. It sounds as if you were showing an integration, but you are actually showing the mean from a water sample integrated from the top 10m. "Dashed lines show the Phases of the experiment as given in Fig. 2," should be moved to the a0 part of the legend, as it is not shown in 3b.

Supplement Figures:

Fig. S2: Top left y axis is formatted differently. Also t vs T as abbreviation for time between S1 and S2

There are two Tables 1 in the supplement.

There is a Fig. S3 that is never mentioned in the text which I suggest actually be moved into the main section as Fig 3c as it shows a difference of DMS/chl among mesocosms!

---

## Referee Comment (RC2) · Anonymous Referee #2 · 12 Apr 2016

**General comment**

This paper presents data from an acidification experiments conducted in large mesocosms in the Baltic Sea during the 2012 summer. The mesocosms system used here has been described in the past and used in previous successful ocean acidification experiments. This is considered as the state-of-the-art system for that type of experiments. As usual in multidisciplinary experiments, many different papers were produced, some of which are already published. This particular paper focuses on the impact of acidification on the production of biogenic trace gases (dimethylsulfide and a suite of halocarbons), but makes several references to other papers related to the same study.

Few general remarks:

1. The upwelling event that took place in the middle of the experiment (t16) certainly confused the issue by cooling the water of the mesocosms. For that reason, the changes in biogenic gases concentrations observed after this event result from both the cooling and the acidification of the water. This is recognized by the authors and properly discussed in this version of the paper.

2. Measurements made outside the mesocosmes are interesting by themselves, and as they are in this version of the paper, should not be compared with the results from the mesocosms where the upwelling event only translated into a decrease in temperature, but no change in salinity and more importantly no change in plankton composition. These are two independent stories which need to be treated as such. In that regard, in situ data could be presented in a separate figure to emphasize this point. A reason to do so is that the Phases indicated in figures 1 and 2 are not relevant to the in situ measurements. This would also allow to rescale the Y-axis of figure 1c and 2a and make the changes in chl-a and DMS concentrations in the mesocosms more visible.

3. The lack of detectable DMSP concentrations is obviously surprising. Although the authors offer possible solutions to this conundrum, the fact remains that they are able to detect a by-product of DMSP degradation but not DMSP itself, known to be, in many circumstances, orders of magnitude higher than DMS. It is difficult to believe that 30 days worth of samples within a diverse community of phytoplankton did not generate a single detectable nmol of DMSP. Some loss can be explained through the presence of acid-sensitive species (colonial Phaeocystis etc.), but the authors rule this out themselves as an important process by specifying that this type of phytoplankton accounted for less than 10% of the community. In fact cryptophytes and chlorophytes dominated the community. Various species of these two groups are known to produce DMSP (Keller et al 1989) but not known to be sensitive to the acid treatment. As stated by the authors, a methodological problem can probably explain these results.
Specific comments

P1, 25: ... challenged Baltic Sea.

P2, 55: ...the global ocean has absorbed... P2,41: Would it be possible to come up with a 'dilution' factor? Using salinity as a conservative parameter perhaps? This would allow to roughly estimate how much of the variability of the parameters measured at the surface needs to be explained by other factors (production/consumption).

P4, 110: Suggestion: replace 'Post-spring bloom' by 'Following the spring bloom'.

P4, 114: ....2012 summer post-bloom season....

P5, 132: ...such as fish...The removal of large zooplankton is probably more relevant here than fish.

P6, 163: ... with 100% absorbance of UV light. ..Later in the manuscript, it is mentioned that some UV light could affect the processes taking place close to the surface in the mesocosms. This seems to be in contradiction that 100% UV is removed.

P8, 230: ....turnover of DMSPD....Replace by 'dissolved DMSP'.

P8, 246: Measurements of carbonate chemistry and community dynamics.

P10, 281: ...decreased over Phase 1 in the ...The phase numbers are not properly aligned in figure 1c (on my printed copy at least), and absent in figure 2, 3 and 4 (which are by the way wrongly numbered).

P10. 287: ... no variation with depth (data not shown)...

P10, 297: ...a significant effect on phytoplankton growth (and biogases production), explaining...

P11, 324: ...that light availability and surface water temperatures...Delete 'environmental conditions of limited' and 'lower'.

P11, 330: A significant 34% reduction...These results could be better explained taking
into account the temporal variability which is significant. Actually, DMS concentrations increased as Chl concentrations decreased, and the increase in DMS was less important at high PCO2. After day 21, DMS decreased gradually in all treatments until the end of the experiment.

P11, 333: (Fig. 3a) to be replaced by (Fig. 2a).

P11, 336: (Fig. 3b) to be replaced by (Fig. 2b).

P11, 337: Furthermore, increases in DMS...were delayed by three days...This 3-day delay is not obvious in Fig. 2a. Am I missing something?

P12, 348: Although the majority...This paragraph needs an introduction sentence. As in my previous review of this paper, I still think that there is too much emphasis on a rare pathway of DMS production considering that the problem is most probably a methodological one. This paragraph is important but could be shortened.

P12, 358: Correlations between...Only one P value is presented. Should it be 'correlation' instead of 'correlations'? I am also wondering if all the data were pooled (all treatments) to compute this statistic.

P12, 373: The peak in DMS concentrations is unlikely to be a delayed response... But the increase in DMS coincided with the decline in Chl-a concentrations (t15-t21), something frequently observed in nature in response to higher DOC production and bacterial activity during bloom decline. My point here is that the results should be presented and discussed in term of temporal changes, not only correlations.

P13, 379: ....2009). DMS and DMSP....

P13, 398: This is relevant. .. I don't understand the logic here. In the absence of DMSP values, whatever the reason, I don't think that one can conclude that 'DMS concentrations were likely more affected by the change in ÆŠCO2 than the production of the precursors'.

BGD
P13, 405: ...and therefore lower DMS microbial yield from DMSP and/or greater consumption of DMS and conversion to DMSO.... DMS yields may vary from 5 to 40% depending on the S and C demand of the bacteria and the quality of DOM. There are many references on variations in DMS yields. A good starting point is the paper by Kiene and Linn 2000 (Distribution and turnover of dissolved DMSP and its relationship with bacterial production and dimethylsulfide in the Gulf of Mexico. Limnol Oceanogr 45: 849-861).

P15, 441: ... where some UV light was able to pass... This seems to be in contradiction with the statement that 100% of UV radiation was absorbed by the cover (P6, 163). This requires clarification.

P15, 455: The peak of CH2I2 coincided with the decline of the bloom, as observed for DMS. I am not convinced that the positive correlations observed between these compounds and the abundance of the different taxa are relevant if the production of the compounds is related to processes linked to the decline of the bloom (ex. increase in DOC).

P15, 466: The cleaning of the walls of the mesocosms and the associated apparent released of DOM as mentioned here seem to be an important potential artifact. As noted, this could be very important for photochemically and microbially driven processes. This potential problem, which could also be important for DMS production, should be discussed in more details in this paper. Would it be useful to indicate on the different figures when these cleanings took place? Overall, providing more details on the impact of these cleaning events would be of great value for colleagues planning to conduct similar long term mesocosms experiments.

P16, 490: ... indicators of algal biomass. PP was not measured here.

P17/177, 503/504: ... low net increase in total Chl-a...

P18, 550: Typo: Two dots before 'but peaked'.
P18, 558: As the CO2 levels increased during Phase II...As mentioned by the authors at the beginning of this section, comparing the mesocosms results with the in situ ones is inappropriate. The different Phases (0, I, II) make only sense for the mesocosms experiment where they indicate either treatments or events. They are irrelevant to the in situ measurements. Keeping this comparison is confusing.

P18, 562: ... this decrease in DMS may also be attributed to CO2 levels....

P19, 577: ... that production was probably not limited...

P19, 598: ...living and acclimated to...

P20, 603-607: These two sentences would benefit from a rewording.

P20, 615: For the concentrations of halocarbons, ...of the Baltic Sea. I am not sure about this conclusion. This is very speculative since deep water upwelling and ocean acidification through air-sea CO2 exchange are two different processes. Upwelling brings nutrients, microbes, etc... in surface water in addition to high CO2.

P 35. This should be Figure 2 (instead of 3).

P 36. This should be Figure 3 (instead of 4).

P 37: This should be Figure 4 (instead of 5).

END

BGD

---

## Author Response (AR1)

[revised manuscript text omitted]

**Reply to Editor Comments**

**1. The lack of DMSP data**

Thanks to you and the reviewers for taking the time to read and edit this manuscript thoroughly. I have made the changes to the manuscript highlighted in the reviewer comments and now include the revised manuscript which outlines the detailed responses we have given to these comments.

The DMSP acidification method is currently used worldwide as a simple and effective method of DMSP storage, and indeed the authors used it in a previous mesocosm experiment with good correlation with samples analysed immediately on a different GC system. However, After the Tvärminne experiment, however, additional tests in this method from the Norfolk broads in varying salinity showed it to be unreliable in some circumstances (but not all), compared to samples which were analysed immediately (Data unpublished). These additional tests, alongside the paper from del Valle et al 2011, showed that this method is unreliable, yet it is still in use, and the authors wanted to highlight these discrepancies, and suggest that significant testing is required prior to heavy reliance on the data generated by this storage method.

With regards to the DMSP issue, in the initial submitted version of the manuscript, there was a significant discussion of why DMSP was not identified during this investigation. This included turnover rates of dissolved DMSP measured in other studies, including consumption by bacteria and phytoplankton, and conversion rates calculated from other studies for DMS. One of the reviewer comments was that this section was very long (which the authors agreed), since we had no measure of turnover or conversion rates, and that we would still see some DMSP even if rates were extremely high, so this section was significantly shortened and put into the methods section. A short discussion of DMS production from methanethiol was included in the results/ discussion (still a potential production route, and one in which we plan further study, hence the need to mention it in this manuscript).

With regards to the reviewer comment you highlighted, this was not addressed directly with a response, as the authors read this as a general statement. Having addressed this reviewer comment throughout the revised manuscript, our response may not have been clear in just the 'response to reviewers' document. Hopefully the revised tracked changed manuscript will show that we have clearly addressed these comments. The discussion of DMS responses from past experiments is quite reliant on the discussion of DMSP, as this is the predominant source of DMS, and the authors believe the discussion of DMSP in the discussion is necessary, assuming that DMSP was present but lost from the samples. I hope that our edited manuscript and this response now clearly explained to you our reasoning behind the decision that this is almost certain a methodological issue. However, if you would like us to clarify anything further please let us know.

**2. Combining mesocosm and Baltic Sea data**

With regards to your comment regarding combining the mesocosm and Baltic Sea data, the authors disagree, and indeed one of the reviewer's comments was that keeping discussions of the mesocosm and the outside waters separate was appropriate, since the outside underwent its own 'experiment'. One of the drawbacks of a mesocosm experiment is that the water is separated from the surrounding environment, in the case of this experiment for over 6 weeks. Given the nature of the water movements through the Storfjärden, after even a few days the phytoplankton population within the mesocosms could potentially be significantly different from the surrounding water masses (dependent on the mesocosm seed population), and therefore the results from the samples within and external to the mesocosms significantly different. This drawback is a point which detractors of mesocosm experiments draw on heavily. As a result, production of trace gases can be significantly different in the mesocosms compared to the outside waters, even if the populations are the same, given that production is based on demands of key elements available in the environment (e.g. sulphur, bromine, iodine) which can change with injections of new water masses. In the case of the halocarbons in particular, macroalgal production is of huge importance to the concentration in the water column, yet macroalgae are not present in the mesocosms, and with a delay of over several days between mesocosm closure and first sampling, sufficient time has elapsed for these gases in the mesocosm water to vent into the atmosphere or break down. The data from the Baltic Sea samples is presented in its own right as an important time series of trace gas analysis, which will add to existing (albeit limited) Baltic trace gas datasets (e.g. Orlikowska & Schulz-Bull 2009; Karlsson et al. 2008; Klick 1992). The change in $CO_2$ and pH in the external waters (to which the mesocosms were not exposed), clearly demonstrates that the community in the Baltic is already adapted to changes in $CO_2$, and therefore helps to explain the lack of change in gas concentrations in the mesocosms. Essentially, although related, measurements from the mesocosms and the Baltic Sea are part of two separate experiments undertaken at the same time, and cannot just be combined easily. It was hoped that this was discussed sufficiently in the final section, but this will be revisited to check if this is so.

The authors would like to thank the reviewer for their comments and discussions at all stages of the review process, which have improved the overall quality of the manuscript. I have addressed the reviewer's comments individually.

1-An initial paragraph or section evaluating the overall quality of the discussion paper ("general comments"),

The manuscript is well structured and for the most parts easily readable. The results show a lack of response of gas concentrations to the experimental design and no linkage to the external conditions due to the outside undergoing its own "experiment", i.e., upwelling. No rates are reported; not clear any were measured. Hence, the entire manuscript must be clarified that the values represent "net" values and not production, nor consumption or degradation, nor emission. Hence, I strongly suggest removing most comments from the discussion that pertain to "climate change" and simply state that concentrations remained the same regardless and emphasize that we (and especially modelers) need to have rates of production, consumption, (photo/chem)-degradation or even "net" rates to include in our prognostic and predictive models.

AR. The mentions of climate change in the discussion section are quite limited, and although the change in halocarbons was limited, the change in DMS concentrations was high between the different treatments. The mention of the Six et al. paper was important, as this study was based on one mesocosm experiment and the results of the model output would have been more significant if the results of a number of mesocosm experiments had been included. A comment has been included to the effect that the reviewer says, that rates of consumption and production of halocarbons are needed to improve model output.

I recommend publishing pending changes. I also suggest shortening some of the longish speculative paragraphs; it's hard to explain why there is no apparent change! In general, I think the manuscript would profit if it was a bit more structured based on hypotheses, rather than being purely descriptive. You must have had some expectations when the experiment was started (and the proposal written!), especially since you had results from previous mesocosm experiments. Especially since no (?) rates were measured.

AR. No rates were measured as it was difficult given the sampling regime of the experiment, without performing additional incubation experiments. There were hypotheses prior to the experiment, mainly that halocarbon concentrations would show some really interesting and varied results under high CO2, as a diazotrophic cyanobacteria bloom occurred. As this bloom did not really occur in the mesocosms, these hypotheses did not apply, particularly since the majority of the 'interesting results' occurred in DMS (but no DMSP). It was therefore important to discuss the lack of DMSP for the community as a whole, to draw to light the issues with the DMSP acidification/ fixing method, and not to concentrate so hard on the lack of changes within the halocarbons.

2- section addressing individual scientific questions/issues ("specific comments"),

The manuscript addresses the influence of ocean acidification on the production of dimethylsulfide (DMS)

and 7 halocarbons in a Baltic Sea mesocosm experiment. The authors effectively found no differences in DMS and halocarbon concentrations over time among the various fCO2 treatments; and no obvious relationship to any other environmental (biological or chemical) variable measured. Difficult to explain without knowing whether turnover is fast. The authors found a decrease of DMS concentrations for highest fCO2 treatments vs. controls only in the last phase (when Chla declined) and none of the other detected differences in halocarbons were CO2 related. The outcome of this study is a relevant piece of information, indicating that most likely there will be no major changes to halocarbon concentrations in the Baltic Sea anytime soon, and the authors conclude that this might be due to the already well adapted community in the unstable Baltic Sea environment with regards to S, T, CO2 and many other factors. The results are interesting by themselves, and valuable for modelers, though modelers need rates. The DMS results again confirm results from a range of mesocosm studies.

3- compact listing of purely technical corrections at the very end ("technical corrections": typing errors, etc.).

Line 247, 248: Inconsistent placing of units, 10m (no space) but 486 nm (space), e.g. line 247, 248 vs 261. You might wanna check if there are more.

AR. Units checked throughout manuscript

Line 70: "Both DMS and DMSP are major routes of sulphur and carbon flux through the marine microbial food web". I wouldn't call them a route, they could be called transporters, or they provide the basis for major routes. Or DMS and DMSP based metabolic pathways are the route…

AR. Changed 'are' to 'provide the basis for'

Line 72: Where do they state that in this reference? I think Simo *et al.*, 2009 should be the reference for phytoplanktonic demand (pages 50-51 e.g.), and Vila Costa *et al.*, 2006a for bacteria (page 653)? Or you put them in as combined references for both (after sulphur demand)

AR. References have been combined at the end of the sentence

Line 142: That was the standard deviation at the beginning? ~50%, ~7%, and ~75%? That's a lot to start with… Any thoughts on how that could potentially affect the outcome of the experiments on the bacterial metabolism side of halocarbon and DMS production?

AR. Certainly for the halocarbons, the difference in nutrients between mesocosms had no effect on the eventual concentrations measured throughout the experiment. Variation in nutrients did not seem to affect DMS concentrations, since the high variation was only identified during the early part of Phase 1, when DMS concentrations did not show differences between mesocosms. Since the differences in DMS were only identifiable during Phase 2, nutrient concentrations had by then showed much lower standard deviation between mesocosms.

Line 169: replace ; with a ,

AR. changed

Line 171: why no comparable pigment analysis, what's the rationale behind it?

AR. Pigment analysis was only carried out in the full 17m depth of the mesocosms. Many other parameters (not discussed in this manuscript) were analysed through the full water depth. In the previous mesocosm experiment, trace gas concentrations were significant in the surface 10m of the water column and diluted by the extra water, so during this experiment, concentrations were taken only from the surface 10m. Flow cytometry was performed both on the surface 10 m and the full 17m depth, but due to a large number of samples, HPLC pigment analysis was only performed on the full water depth.

Line 179: Is that shown anywhere? Otherwise please state what the precision was, and that it is not further shown.

AR. Precision calculated as the percent deviation and inserted in manuscript.

Line 237: careful. The del Valle et al samples were DMSPd and DMSPt; not DMSPp. The Kiene group estimates DMSPp by difference between the total and dissolved pool.

AR. It is uncertain what the reviewer is highlighting here. The samples during the mesocosm experiment were DMSPt and DMSPp, but none of them showed any DMSP within. In the manuscript, there is no distinct emphasis on DMSPp over DMSPt, as the reviewer seems to be suggesting there was. This has been clarified in the manuscript by using DMSPt instead of just DMSP.

Line 247: 17mIWS space needed

AR. changed

Line 248: Chl-a the a is superscripted

AR. Could not find this error. Most likely corrected already.

Line 300: Mixing of the mesocosms after closure prior to t-3 did not trigger a notable increase in Chl-α in Phase 1; in previous mesocosm experiments, mixing redistributed nutrients from the deeper stratified layers throughout the water column

I get what you are saying, but I think you should add what redistributing nutrients did- I am assuming here that it lead to an increase in Chl-a?

AR. In previous mesocosm experiments, redistribution of the nutrients from below the stratified surface layers results in a significant bloom of Chl-a. However, this was not identified during this experiment, suggesting a limiting factor. The manuscript has been clarified on this point.

Line 282: "mainly through air-sea gas exchange" – isn't that usually considered to be limited by the small surface area / volume ratio? Please comment on why this should not be the same for your analyzed gases.

AR. The sea-air exchange will still exist during the mesocosm experiment, but it will be significantly decreased due to restrictions on wind interactions due to the mesocosm walls, reduced wave action and a very low SA: vol ratio for the water in the mesocosm. It is acknowledged that the trace gases will be lost to the atmosphere in the same way as $CO_2$, but at different rates for different compounds. This is commented on later on particularly for the bromocarbons which showed a steady decrease in concentration throughout the experiment. We know that there was a steep $CO_2$ gradient to the atmosphere, we do not know the concentration gradients for the halocarbons or DMS: if atmospheric halocarbon concentrations are equivalent to concentrations in the mesocosms (potentially possibly in the forests of finland), there would be a significantly lower rate of halocarbon loss than $CO_2$ loss from the mesocosms. Existing flux models cannot be used due to the constraints of the mesocosm enclosure.

Line 302: no direct result of the CO2 additions because there was no significant difference between controls and treatments?

AR. Sentence clarified with 'as no difference was identified between enriched mesocosms and controls'

Line 309: chlorophytes (largest contributor to chl a) are not exactly known to be high DMSP or DMS producers; you may want to mention that given stated link to pico and nanoeukaryotes as possible sources.

This is why bringing in the Fig, S3 as Fig 3c somewhere actually shows that there are differences among treatments.

AR. L 382, a statement was added to the DMS and community parameters section stating 'Of the studied phytoplankton groupings, neither the cryptophyes or chlorophyes as the largest contributors of Chl-a have ever been identified as significant producers of DMSP.'

It was decided to keep figure S3 in the supplementary, as DMS is clearly disconnected from total Chl-a concentrations, during Phase 2, which is not due to changes in Chl-a. The statistics on the DMS: Chl-a ratio are also insignificant due to the high standard deviation. This plot was therefore given for interest in the supplemental, but was not considered sufficiently robust to include in the finished manuscript.

Line 311-312: so between the opposing trends for pico I and pico II, the next effect on DMS in the system is zero?

AR. This section is discussing the differences seen in the mesocosms, and is not discussing the DMS concentrations. It is not implicitly stated that these groups are directly responsible for the DMS concentrations. There are many parameters acting on the DMS concentrations on top of the changes in these groups.

Line 331: Please explain F-test or at least the H0 you used in one sentence in the methods section.

AR. Null hypothesis added to the methods section

Line 348-369: Simply there was no relationship between patterns (or lack thereof) in DMS concentrations and any other measured variable. And no rate measurements available. Please say so. Too many possibilities, too many unknowns. This section reads a bit like "filler"; sorry.

AR. This section was significantly reduced prior to online discussion. It was decided that this section was necessary to discuss the alternate production pathways of DMS that could potentially be available in the Baltic Sea. A discussion occurred as to how this could be further investigated, and the authors felt it was important to keep this section relatively whole to allow for further research into these pathways, with this manuscript as a starting point.

Line 354: synthesis should be synthesise

AR. changed

Line 358: Correlations between DMS and the cyanobacterial equivalent Chl-α 359 ($p$=0.42, p<0.01) indicate that the methylation pathway may be a potential source of DMS within the 360 Baltic Sea community. Reference? Data shown anywhere?

AR. The cyanobacterial equivalent Chl-a and the single celled cyanobacterial abundance are shown in Supplementary. They were previously included in the paper, but it was considered too much data for too little solid evidence. This sentence therefore keeps the idea that there MAY be a relationship between DMS and cyanobacterial activity, but does not outright state that there is. The authors feel this could be a significant area of research that needs further investigation.

Line 367: Stop! What rates of net DMS production? Did you measured or estimate them? If you did, please indicate and discuss!

AR. No rates of production were calculated, hence the 'net' DMS production (concentrations remain the same despite removal and addition processes). However, in this instance this has been changed to 'measured DMS concentrations'.

Line 371: but I thought that Syn does not make DMS?! There never is high DMS concentration reported along with it in subtrop regions (DiTullio et al., others). Didn't Vla-Costa et al. 2006 report uptake of DMSPd (not DMS) by Syn and other picoeukaryotes?

AR. Other literature has not identified Syn as a significant producer of DMS or DMSP. This statistics reported a significant correlation with cyanobacteria, which is reported here, both the single-celled and multi-celled variety (Data not shown). This section has been amended by the addition of 'predominantly' Synechococcus' as it is likely there are other single celled cyanobacteria within the population aside from Syn.

Line 372: Why is it unlikely?

AR. It has never been observed previously that a DMS peak occurs 5 days after a peak in Chl-a which has been directly linked to the Chla peak. DMS and Chla concentrations are rarely coupled, indeed even DMSP is rarely coupled to Chla, so this result is not unexpected.

Line 379: just one period.

AR. removed

Line 386: "However, these experiments limit our ability to generalize"… I don't think it's the experiments limiting, but rather the varying responses, is that what you are saying?

AR. Essentially, yes. The mesocosm experiments have been measuring DMS, DMSP and community parameters for a number of years now, and yet still no consensus appears as to the response to community changes. Mesocosm experiments also have their distinct disadvantages in their own right. This sentence has been amended to 'the varying response within the mesocosm experiments'

Line 410-411: no data on consumption, no bacterial rates described, then what is the basis for this statement? Confusing.

AR. This statement has been amended to 'it is not known if this loss pathway is stimulated at high CO2'

Line 412: "Synechococcus has been identified 412 as a DMS consumer in the open ocean" Reference, please. Syn consumed labelled DMSPd, not DMS (Vila-Costa et al 2006)

This reference to Syn has been removed, as it is a DMSP consumer, not DMS, as the reviewer states.

Line 431: Sections 3.3 and 3.4: No rates of anything for the halogenated compounds either? Just checking.

AR. No, no rates were measured, due to the sampling limitations of the mesocosm experiment.

Lines 518-522: well, was the region isolated from the coastal environment or not? You can't have it both ways. I understand that the mesocosm bags were closed so they wouldn't have a macroalgal component. This will come back in the discussion

AR. The water within the bags was isolated from the outside environment, but this statement was to highlight that halocarbons were likely present in high concentrations in the water column prior to the mesocosm installation and closure. This would therefore have influenced halocarbon, particularly bromocarbon concentrations at the beginning of the experiment. This section has been reworded to make this clearer.

Line 548: I agree that the comparison between the mesocosms and the outside is inappropriate. The outside underwent its own and different "experiment"

Line 557: please delete sentence about DMSp as it implies that there was none because none detected when it is an analytical issue

AR. removed

Line 558-569: given the statement in Line 548, please remove this paragraph as it mixes mesocosm conditions with outside conditions. It is pure speculation as a lot more changed with the injection of upwelled water than fCO2- i.e., particles, nutrients, DOM, etc, etc

AR. Paragraph deleted.

Line 576: Is CH2ClI really polyiodinated?

AR. polyhalogenated

Line 584: Check your manuscript for Chl-α, the α is alternating between superscript and normal

AR. Checked – all changed to italic

Line 586-590: It is above indicated that macroalgal beds were not a source. Now, it is implied that those macroalgals beds were close? or far? in location w/r to the mesocosms. And the prevailing circulation was from the beds towards the mesocosms? And waht about vertical input? The entire DMS section is predicated on upwelling, ie, water injection from below NOT lateral advection. Can't have tvertical input for one gas and horizontal input for the other one.

AR. The mesocosms were approximately 500m from the shore, however the maximum depth of the seabed was 20-25m, so macroalgae growing there would have been within a few metres of the water taken from the Baltic. There was also free-floating macroalgae in the water column which could have contributed.

The mesocosms were set up in a Fjard, which although had minimal tidal impact, had obvious signs of water movement in and out, with significant currents identified when mooring the boats to the mesocosms for sampling.

A comment was included which stated that there was limited change in bromocarbons during the upwelling, likely that the upwelled water had similar concentrations to the surface waters.

Line 593-607: good

Line 599: I think you want to stress here, that the values are high enough to be considered an already adapted site, rather than stressing that they are lower than elsewhere, correct? "[…] at such a location with a relatively low fCO2 excursion compared to some sites […]", maybe rephrase to "[…] at such a location with a relatively high fCO2 excursion, however still relatively low when compared to some sites […]"

AR. Agreed and changed.

Line 609-611: Not all the time, only after the decline of Chl-a, right? I wouldn't stretch it out, then.

AR. This statement was included as it was the most important finding of the mesocosm experiment, and compared to all the other mesocosm experiments.

Line 614: production was not measured, only concentrations. Please change production for cycling because the levels measured are a net result

AR. Changed

Line 615: since rates were not measured, you don't know whether was a response (ie, prodn and/or cons), only that the measured concentrations did not change

AR. Changed to 'the measured concentrations did not change'

Line 617: no change IF under similar meteorological conditions as during this sampling

AR. Added the proviso 'without significant alteration top the meteorological conditions'

Line 617-621: NET production or availability. Again, same issue. Also, rather simplistic as meteorology must be considered.

AR. Added 'net'

L621-625: This is a weak concluding paragraph. It says nothing at all. Keep it honest and simple by saying that no changes in concentrations were seen and that next time it would be best to measure rates so these rates can be included in models to have better predictions!! So sorry that you didn't see any changes nor anything "exciting".

AR. This paragraph has been amended to include the reviewers comments.

Figures in general:

I find it very irritating how the units are given, e.g. fCO2/µatm. I read the "/" as 'per', which makes it confusing. I would very much prefer if you put fCO2 (µatm) or fCO2 [µatm]

AR. The figures have been amended to have the units in brackets

Fig 3: The Legend is misleading. It sounds as if you were showing an integration, but you are actually showing the mean from a water sample integrated from the top 10m. "Dashed lines show the Phases of the experiment as given in Fig. 2," should be moved to the a0 part of the legend, as it is not shown in 3b.

AR. Legend amended

Supplement Figures:

Fig. S2: Top left y axis is formatted differently. Also t vs T as abbreviation for time between S1 and S2

AR. Figures have been amended.

There are two Tables 1 in the supplement.

AR. Table S2 renamed

There is a Fig. S3 that is never mentioned in the text which I suggest actually be moved into the main section as Fig 3c as it shows a difference of DMS/chl among mesocosms!

AR. The standard deviation during this figure is so high that it is a non-significant finding, and it has been established that there is no link between DMS and Chla. This figure was originally in the manuscript but was removed during the first round of reviewer comments prior to online discussion.

The authors would like to thank the reviewer for their comments and discussions at all stages of the review process, which have improved the overall quality of the manuscript. I have addressed the reviewer's comments individually.

General comment This paper presents data from an acidification experiments conducted in large mesocosms in the Baltic Sea during the 2012 summer. The mesocosms system used here has been described in the past and used in previous successful ocean acidification experiments. This is considered as the state-of-the-art system for that type of experiments. As usual in multidisciplinary experiments, many different papers were produced, some of which are already published. This particular paper focuses on the impact of acidification on the production of biogenic trace gases (dimethylsulfide and a suite of halocarbons), but makes several references to other papers related to the same study.

Few general remarks:

1. The upwelling event that took place in the middle of the experiment (t16) certainly confused the issue by cooling the water of the mesocosms. For that reason, the changes in biogenic gases concentrations observed after this event result from both the cooling and the acidification of the water. This is recognized by the authors and properly discussed in this version of the paper.

2. Measurements made outside the mesocosms are interesting by themselves, and as they are in this version of the paper, should not be compared with the results from the mesocosms where the upwelling event only translated into a decrease in temperature, but no change in salinity and more importantly no change in plankton composition. These are two independent stories which need to be treated as such. In that regard, in situ data could be presented in a separate figure to emphasize this point. A reason to do so is that the Phases indicated in figures 1 and 2 are not relevant to the in situ measurements. This would also allow to rescale the Y-axis of figure 1c and 2a and make the changes in chl-a and DMS concentrations in the mesocosms more visible.

AR. This has previously been discussed, however it doubled the number of figures in the manuscript, while not increasing the clarity of the information displayed to a huge degree. The differences in DMS concentration in the different mesocosms is clearly visible in the current figure 3 due to the scale of the difference.

3. The lack of detectable DMSP concentrations is obviously surprising. Although the authors offer possible solutions to this conundrum, the fact remains that they are able to detect a by-product of DMSP degradation but not DMSP itself, known to be, in many circumstances, orders of magnitude higher than DMS. It is difficult to believe that 30 days worth of samples within a diverse community of phytoplankton did not generate a single detectable nmol of DMSP. Some loss can be explained through the presence of acid-sensitive species (colonial Phaeocystis etc.), but the authors rule this out themselves as an important process by specifying that this type of phytoplankton accounted for less than 10% of the community. In fact cryptophytes and chlorophytes dominated the community. Various species of these two groups are known to produce DMSP (Keller et al 1989) but not known to be sensitive to the acid treatment. As stated by the authors, a methodological problem can probably explain these results.

Specific comments
P1, 25: . . .challenged Baltic Sea.

AR. Challenging is more appropriate as the sentence is talking about the challenges present and future in the Baltic Sea encountered by phytoplankton.

P2, 55: . . .the global ocean has absorbed. . .

AR. Changed

P2,41: Would it be possible to come up with a 'dilution' factor? Using salinity as a conservative parameter perhaps? This would allow to roughly estimate how much of the variability of the parameters measured at the surface needs to be explained by other factors (production/consumption).

AR. We do not know the salinity of the upwelling water, nor the percentage volume of the upwelled water injected into the surface system. This makes this very hard to quantify.

P4, 110: Suggestion: replace 'Post-spring bloom' by 'Following the spring bloom'.

AR. Changed

P4, 114: . . .2012 summer post-bloom season. . .

AR. Changed

P5, 132: . . .such as fish. . .The removal of large zooplankton is probably more relevant here than fish.

AR. Although it took a week to get 1 small fish out…

P6, 163: . . .with 100% absorbance of UV light. . .Later in the manuscript, it is mentioned that some UV light could affect the processes taking place close to the surface in the mesocosms. This seems to be in contradiction that 100% UV is removed.

AR. UV was still able to impact the very surface waters where it did not pass through the films. There is a 1m high gap in the mesocosm design between the top of the TPU bag and the PVC rain cover, where the samples are taken from. Light is able to pass through this gap in morning and evening and hit the surface waters.

P8, 230: . . .turnover of DMSPD. . .Replace by 'dissolved DMSP'.

AR. Changed

P8, 246: Measurements of carbonate chemistry and community dynamics.

AR. Changed

P10, 281: . . .decreased over Phase 1 in the . . .The phase numbers are not properly aligned in figure 1c (on my printed copy at least), and absent in figure 2, 3 and 4 (which are by the way wrongly numbered).

AR. Figures have been amended

P10. 287: . . .no variation with depth (data not shown). . .

AR. Added

P10, 297: . . .a significant effect on phytoplankton growth (and biogases production), explaining. . .

AR. Added

P11, 324: . . .that light availability and surface water temperatures. . .Delete 'environmental conditions of limited' and 'lower'.

AR. Agreed

P11, 330: A significant 34% reduction. . .These results could be better explained taking into account the temporal variability which is significant. Actually, DMS concentrations increased as Chl concentrations decreased, and the increase in DMS was less important at high PCO2. After day 21, DMS decreased gradually in all treatments until the end of the experiment.

AR. The DMS disconnect from Chl-a is a fairly common occurrence, and it would have been a lot more interesting to discuss if DMS had been connected to Chl-a concentrations! To a degree, it is interesting that DMS peaked after the Chl-a, but without any DMSP measurements, it is difficult to know to what degree this was connected. From previous mesocosm experiments and turnover rates of DMS, the temporal delay in DMS peak after Chl-a (if it exists) is usually only 2-3 days, not over a week.

P11, 333: (Fig. 3a) to be replaced by (Fig. 2a). P11, 336: (Fig. 3b) to be replaced by (Fig. 2b).

AR. Changed

P11, 337: Furthermore, increases in DMS. . .were delayed by three days. . .This 3-day delay is not obvious in Fig. 2a. Am I missing something?

AR. The increase in DMS in the highest CO2 mesocosms started three days after that in the ambient and mid-level CO2. As the DMS increased to such a small degree in the high CO2, it is not an obvious result, however it can be seen in Fig. 2.

P12, 348: Although the majority. . .This paragraph needs an introduction sentence. As in my previous review of this paper, I still think that there is too much emphasis on a rare pathway of DMS production considering that the problem is most probably a methodological one. This paragraph is important but could be shortened.

AR. The first sentence has been amended to be more of an introduction. This paragraph has been shortened significantly from the original version, and to shorten it further would be to miss out the summary of where knowledge of the alternate pathway originates from and how it affects the results of this experiment.

P12, 358: Correlations between. . .Only one P value is presented. Should it be 'correlation' instead of 'correlations'? I am also wondering if all the data were pooled (all treatments) to compute this statistic.

AR. There was also correlation between the single celled cyanobacterial abundance, which has been included, and the colonial cyanobacterial abundance (data not shown as not finalised when preparing the manuscript). The statistics are also given in the supplemental file.

P12, 373: The peak in DMS concentrations is unlikely to be a delayed response. But the increase in DMS coincided with the decline in Chl-a concentrations (t15-t21), something frequently observed in nature in response to higher DOC production and bacterial activity during bloom decline. My point here is that the results should be presented and discussed in term of temporal changes, not only correlations.

AR. Comments have been included as to the temporal variation in DMS concentrations between the mesocosms, and as mentioned above, it is not uncommon for there to be a complete disconnect between Chl-a and DMS, and we have no DMSP concentrations to form a connection between the two. There was an increased in DOC on t15 shortly before the DMS peak, which has been referenced to Hornick et al 2016 (this issue).

P13, 379: . . .2009). DMS and DMSP. . .

AR. Changed

P13, 398: This is relevant. . .I don't understand the logic here. In the absence of DMSP values, whatever the reason, I don't think that one can conclude that 'DMS concentrations were likely more affected by the change in ÆŠCO2 than the production of the precursors'.

AR. Final sentence deleted

P13, 405: and therefore lower DMS microbial yield from DMSP and/or greater consumption of DMS and conversion to DMSO. DMS yields may vary from 5 to 40% depending on the S and C demand of the bacteria and the quality of DOM. There are many references on variations in DMS yields. A good starting point is the paper by Kiene and Linn 2000 (Distribution and turnover of dissolved DMSP and its relationship with bacterial production and dimethylsulfide in the Gulf of Mexico. Limnol Oceanogr 45: 849-861).

AR. A comment has been included to the effect that bacterial consumption varies to a wide degree.

P15, 441: . . .where some UV light was able to pass. . .This seems to be in contradiction with the statement that 100% of UV radiation was absorbed by the cover (P6, 163). This requires clarification.

AR. See comment above

P15, 455: The peak of CH2I2 coincided with the decline of the bloom, as observed for DMS. I am not convinced that the positive correlations observed between these compounds and the abundance of the different taxa are relevant if the production of the compounds is related to processes linked to the decline of the bloom (ex. increase in DOC).

AR. There is no direct evidence of a link between the production of these compounds, but there is also no evidence that this link does not exist. This is why this is presented as a correlation, but does not equal causation, and was not described as such here.

P15, 466: The cleaning of the walls of the mesocosms and the associated apparent released of DOM as mentioned here seem to be an important potential artifact. As noted, this could be very important for photochemically and microbially driven processes. This potential problem, which could also be important for DMS production, should be discussed in more details in this paper. Would it be useful to indicate on the different figures when these cleanings took place? Overall, providing more details on the impact of these cleaning events would be of great value for colleagues planning to conduct similar long term mesocosms experiments.

AR. Cleaning during the experiments was not as regular as was hoped for, and only took place during the second part of the experiment. Because of this it is likely that the cleaning had a significant effect on DMS concentrations due to the input of DOC into the mesocosm. A comment to this effect has been included.

P16, 490: . . . indicators of algal biomass. PP was not measured here.

AR. Changed

P17/177, 503/504: . . .low net increase in total Chl-a. . .

AR. Added

P18, 550: Typo: Two dots before 'but peaked'.

AR. Removed

P18, 558: As the CO2 levels increased during Phase II. . .As mentioned by the authors at the beginning of this section, comparing the mesocosms results with the in situ ones is inappropriate. The different Phases (0, I, II) make only sense for the mesocosms experiment where they indicate either treatments or events. They are irrelevant to the in situ measurements. Keeping this comparison is confusing.

AR. A bit of the comparison is removed. The phase has been changed to the day no.

P18, 562: . . .this decrease in DMS may also be attributed to CO2 levels. . ..

AR. Section removed

P19, 577: . . .that production was probably not limited. . .

AR. Changed

P19, 598: . . .living and acclimated to. . .

AR. Changed

P20, 603-607: These two sentences would benefit from a rewording.

AR. Last sentence has been restructured.

P20, 615: For the concentrations of halocarbons, . . .of the Baltic Sea. I am not sure about this conclusion. This is very speculative since deep water upwelling and ocean acidification through air-sea CO2 exchange are two different processes. Upwelling brings nutrients, microbes, etc. . . in surface water in addition to high CO2.

AR. This section has been reworded.

P 35. This should be Figure 2 (instead of 3).

AR. Changed

P 36. This should be Figure 3 (instead of 4).

AR. Changed

 P 37: This should be Figure 4 (instead of 5).

AR. changed

---

## Referee Report (RR1)

Review of "Effect of ocean acidification and elevated fCO2 on trace gas production by a Baltic Sea summer phytoplankton community" Author(s): A.L. Webb et al. MS No.: bg-2015-573

**1-An initial paragraph or section evaluating the overall quality of the discussion paper ("general comments"),**

The manuscript quality has improved since the last review and the reviewer comments were addressed adequately. The manuscript is much more concise and less speculative.

The results show a lack of response of gas concentrations to the experimental design and no linkage to the external conditions due to the outside undergoing its own "experiment", i.e., upwelling.

No rates were measured, which has been clarified, concentrations (mostly) remained the same, and the need to have rates of production, consumption, (photo/chem)-degradation or even "net" rates to include in prognostic and predictive models has been emphasized more.

I again recommend publishing pending changes. Most of the (many-sorry!) small corrections (see below) are easily done. For me personally it would be important to have a clearer number agreement for the mean  $CO_2$  values (see comment on Line 29). Also I think the firm and clearly stated replies to the editor comments under Point 1 and 2 would strengthen the discussion, as they nicely address the criticism that could still emerge. It could be worth to have a (short, taken from the comments) discussion section on methodological caveats/challenges.

**2- section addressing individual scientific questions/issues ("specific comments"),**

Unchanged: The manuscript addresses the influence of ocean acidification on the production of dimethylsulfide (DMS) and 7 halocarbons in a Baltic Sea mesocosm experiment.

The authors effectively found no differences in DMS and halocarbon concentrations over time among the various fCO2 treatments; and no obvious relationship to any other environmental (biological or chemical) variable measured. Difficult to explain without knowing whether turnover is fast. The authors found a decrease of DMS concentrations for highest fCO2 treatments vs. controls only in the last phase (when Chl-a declined) and none of the other detected differences in halocarbons were CO2 related.

The outcome of this study is a relevant piece of information, indicating that most likely there will be no major changes to halocarbon concentrations in the Baltic Sea anytime soon, and the authors conclude that this might be due to the already well adapted community in the unstable Baltic Sea environment with regards to S, T,  $CO_2$  and many other factors. The results are interesting by themselves, and valuable for modelers, though modelers need rates.

The DMS results again confirm results from a range of mesocosm studies.

3- ompact listing of purely technical corrections at the very end ("technical corrections": typing errors, etc.).

Comment upfront: I didn't actively look for any typing errors et cetera at this state of the manuscript, these are just the things that "jumped at me".

Line 25: additional stressor facing the pelagic community of the already challenging Baltic Sea. Doesn't the community have to face the stressor? I tried to find this formulation anywhere else, but failed.

Line 29: I am confused by the mean CO2 concentrations given. Here it is  $350 \ \mu atm$  (which I would read as a rounded value if the highest wasn't given as 1333, so as the correct number). In Table 1 the two means are 331 and 334 (= ~ 332-333 if combined). In Figs. 1, 2, 3, 4 and S1, S2 they are 346 and 348 (=~347 if mean of both). In S3 the mean is 346.

Line 104: separates CO2-rich, bottom waters from fresher, lower..., after  $CO_2$ -rich? Line 49-50: "however emissions of biogenic sulphur could significantly decrease from this region" sounds strange to me – (no native speaker, though). Shouldn't it read: however emissions of biogenic sulphur from this region could significantly decrease. Nitpicking. Line 111-112: are largely unstudied in 112 terms of their community trace gas production during the **summer bloom**. You introduce a knowledge gap during the bloom (which you expected to happen) and then (line 117-118) state that you "report the concentrations of DMS, DMSP and halocarbons from the 2012 summer **post-bloom season** mesocosm experiment". I think you should match these two.

Line 113-114: "a low dissolved inorganic nitrogen (DIN) to dissolved inorganic phosphorous 114 (DIP) ratio combines with high temperatures and light intensities to encourage" I get what you're saying, but I am fairly sure they don't combine TO encourage, but "combined with... **encourages**" or ", encouraging" instead of "to encourage"?

Line 184: below 10°C in the dark: you have a space in between everywhere else.

Line 216: 7 mL sample vial, really? Or 8 mL as the one before from Labhut?

Line 219: purged with 1 mL of 10M NaOH for 5 minutes at 80 mL. Pretty sure you didn't purge WITH NaOH, but you purged after NaOH addition with ??? right? OFN maybe?

Line 262: in 90 % acetone with: NO space before % or everywhere else

Line 274-276: In analysis of the measurements **between mesocosms**, one-way ANOVA was used with Tukey's post-hoc analysis test to determine the effect of different *f*CO2 on concentrations measured **in the mesocosms and the Baltic Sea** – with or without the Baltic?

Line 317-318: this decrease was attributed to a temperature induced decreased in

Line 363: 7).Recent studies: Space before Region.

Line 383-384: "**neither** the cryptophyes **or** chlorophyes as the largest contributors of Chl-a **were** identified as significant producers of DMSP." Neither...**n**or, and during this study? Or **have been** identified – before and general?

Line 392-395: "The variation 393 in inorganic nutrient concentrations between mesocosms at the start of the experiment did not have 394 an effect on DMS concentrations during Phase I, and by the start of Phase II the variation between 395 mesocosms had decreased." If they started at varying levels and then were more similar, doesn't that indicate a different usage in the different mesocosms? And wouldn't that imply a different effect on the respective communities then? Impact on later development?

Line 462: "surface **10m** of the" Space please.

Line 487: "However, given the **lack of response** of these compounds to elevated fCO2 (F=1.7, **p<0.01**), it" p<0.01 = lack of response? Confused.

Line 494-496: "Hughes et al. (2008) did not identify this route as a pathway for 495 CH2I2 or CH2CII production, but Carpenter et al. (2005) suggested a production pathway for these 496 compounds through the reaction of HOI with aggregated organic materials." This sentence seems a little lost here. Maybe a concluding sentence would help, even if it is (again) just a statement that you (unfortunately) can't solve this due to a lack of measurements on these routes/compounds?

Line 508-509: "decreased steadily in all mesocosms from t-3 through to t31, over the range 4.0 to 7.7 pmol  $L^{-1}$ ". It decreased over a range sounds a little awkward to me.

Line 519-524: "Production of all three bromocarbons was identified from large-size cyanobacteria such as Aphanizomenon flos-aquae by Karlsson et al. (2008), and in addition, significant correlations were found in the Arabian Sea between the abundance of the cyanobacterium Trichodesmium and several bromocarbons (Roy et al., 2011), and the low abundance of such bacteria in the mesocosms would explain the low

variation in bromocarbon concentrations through the experiment." That is one long sentence. Why not split it after (Roy et al., **2011). The** low abundance...

Line 526: "suggested **as** of greater importance than" Again, I am no native speaker- this sounds weird to me: "suggested **to be** of greater importance **than**" or "suggested **to be** more important **than**"?

Line 528: "growth rates and low net increase in total Chl-a" during the experiment described herein or the like?

Line 539: "a greater concentrations gradient" change to "a greater concentration gradient"

Line 530: "bacterial breakdown; which could explain" comma or this

Line 555: the associated pH, as well as having communities associated with the

Line 576: L-1**b**ut peaked: Space.

Line 578: I still couldn't find a reference to Fig S3, here seems about right? "ratio of DMS: Chl-a at 1.6 ( $\pm$  0.3) nmol  $\mu g^{-1}$  (Fig. S3)"

Line 583: after t17the DMS. The last time I write the word space...

Line 586: "DMS deep water" Period please.

Line 600: "with maximum concentrations 191.6 pmol L-1, 10.0 pmol L-1 and 5.0 pmol L-1 respectively" with maximum concentrations of 191.6 pmol L-1, 10.0 pmol L-1 and 5.0 pmol L $^{-1}$ , respectively

Line 647: mesocosm, field, and laboratory

Tables:

Table 1: All  $CO_2$  means given here differ from the ones given throughout the rest of the manuscript. I am guessing that this might be due to the inclusion of the t- days? If so, what is the rationale in having the days before  $CO_2$  addition (t0) in the averages for the  $CO_2$  treatments? I would start with t0. Again, I personally find it very irritating if I find different numbers used. And: The Baltic had a target (?) but the controls not (=Baltic?)

Table 3: I like the information given here, however I do not like the table very much. It has a lot of empty space. And Chl a has 100% contribution to Chl a? (2) I know this kind of table is hard to make nice, but maybe you could restructure it... Or at least fill some of that space by indicating ND for the taxonomy in the 10 m integrated samples? Use some abbreviations to get narrower columns? And isn't it still mesocosms?

Table 4: Again, I like the Information- and in this case also the table. Trying to do any comparisons from these ranges in controls and highest fCO2 treatments is really hard, though. Hmm, It would be great to have sort of a standard case, like 750 vs. control or the like, but I see that it would be a lot of work/calculations given all the compounds and experiments. Not insisting on you changing it, just saying.

Figures:

Fig. 1: Missing: fCO2 shown in the legend are mean fCO2 across the duration of the experiment (you have it in all other legends).

Supplement Figures:

Fig. S1: "mesocosm<mark>s D</mark>ashed" Add period

Fig. S3: "Paul *et al*. (2015)" Add period.

---

## Author Response (AR2)

Thank you very much to the reviewers and editors for their time on this manuscript. The authors have reviewed the changes suggested in the technical corrections, and have highlighted the changes here. A tracked changes version of the manuscript is included.

Reviewer 1

Line numbers corresponding to the document still including the changes.
Line 93: …annual flux….

Changed

Line 396: 't' needs to be in italics.

Changed

Line 397: …as well as a response to the mesocosm wall cleaning which took place on t16…this part of the sentence does not fit with the beginning of the sentence. Needs to be reworded.

The section about wall cleaning has been put into a second sentence. Now reads 'These higher DMS concentrations were likely connected to a peak in dissolved organic carbon (DOC) on $t$15, as well as increasing bacterial abundance during Phase II (Hornick *et al.*, 2016). It is also likely that DMS concentrations increased as a response to the mesocosm wall cleaning which took place on $t$16.'

Line 441: …vary…Typo changed

Line 544: …coastal environment. However, …

changed

Line 599: …t17 or t17 as before and just one line below?

CO2 did increase between t16 and t17, but the largest increase was seen after t17. However, as the first increase was seen after t16, and for ease of discussion, this has been changed to t16, as it corresponds with the change in DMS.

Line 651: …hard… I don't think that 'hard' is the appropriate word to use here.

Suggestion: …therefore these upwelling events cannot be considered as natural high CO2 analogues. But the beginning of this sentence needs to be reworded also.

The section has been reworded to 'The upwelling event occurring mid-way through our experiment allowed comparison of the mesocosm findings with a natural analogue of the system, as well as showing the extent to which the system perturbation can occur (up to 800 $\mu$atm). This event was a fortuitous occurrence during this mesocosm experiment, but as the scale and timing of these upwelling events is difficult to determine, and therefore these upwelling events are extremely challenging to study as natural high $CO_2$ analogues. '

Reviewer 2

The data in this manuscript are important information.
The results show a lack of response of gas concentrations to the experimental design and no linkage to the external conditions due to the outside undergoing its own "experiment", i.e., upwelling. No rates were measured, concentrations (mostly) remained the same, and the need to have rates of production, consumption, (photo/chem)-degradation or even "net" rates to include in prognostic and predictive models has been emphasized.

I recommend publishing pending changes. For me personally it would be important to have a clearer number agreement for the mean CO2 values.

Could the reviewer clarify what is meant by this, as I do not understand?

Also I think the firm and clearly stated replies to the editor comments under Point 1 and 2 would strengthen the discussion, as they nicely address the criticism that could still emerge.

The section on DMSP has been edited to include the following (TC version)

L 235 This method had been used during a previous mesocosm experiment (Bergen, Norway) and the results correlated well with those measured immediately on a similar GC-FPD system (Webb *et al.* 2015).

L 255 The DMSP acidification method is used worldwide as a simple and effective method of DMSP storage. The findings here, alongside those of del Valle *et al.* (2011) question the applicability of this method in other marine environments, and suggests significant testing prior to reliance on this method as a sole means of DMSP storage.

The section on the Baltic Sea has been amended to include comments from Section 2 of the replies to editors. Specifically (TC version):

L 578 The changes in biological parameters and trace gas concentrations are therefore discussed here separately from the concentrations measured in the mesocosms.

L 581 Given the separation of the waters within the mesocosms, and the movement of water masses within the Baltic Sea, it is expected that phytoplankton population structure could be significantly different inside the mesocosms compared to the external waters.

It could be worth to have a (short, taken from the comments) discussion section on methodological caveats/challenges and reasoning for the way the data is presented.

In terms of this, I believe this comment has been covered in the individual sections mentioned, and to include this recommended section would cause repetition within the manuscript. It is fair to say that the layout of the sections has caused a lot of discussion during the preparation of this manuscript.

L 607 In addition, the community demands of sulphur are likely to be very different in the Baltic Sea compared to the mesocosms, due to differences in community composition and sulphur availability, and therefore direct comparisons with mesocosm concentrations are inappropriate.

L634 Macroalgal production in the Baltic Sea is likely the predominant iodocarbon source, compared to the mesocosms where macroalgae are excluded.

**Effect of ocean acidification and elevated $f$CO$_2$ on trace gas production by a Baltic Sea summer phytoplankton community**

**A.L. Webb[1,2], E. Leedham-Elvidge[1], C. Hughes[3], F.E. Hopkins[4], G. Malin[1], L.T. Bach[5], K. Schulz[6], K. Crawfurd[7], C.P.D. Brussaard[7,8], A. Stuhr[5], U. Riebesell[5], and P.S. Liss[1].**

[1] {Centre for Ocean and Atmospheric Sciences, School of Environmental Science, University of East Anglia, Norwich, UK, NR4 7TJ}

[2] {Groningen Institute for Evolutionary Life Sciences, University of Groningen, 9700 CC Groningen, The Netherlands}

[3] {Environmental Department, University of York, York, UK, YO10 5DD}

[4] {Plymouth Marine Laboratory, Plymouth, UK, PL1 3DH}

[5] {GEOMAR Helmholtz Centre for Ocean Research Kiel, Düsternbrooker Weg 20, 24148 Kiel, Germany.}

[6] {Centre for Coastal Biogeochemistry, School of Environment, Science and Engineering, Southern Cross University, Lismore, NSW 2480, Australia.}

[7] {Department of Biological Oceanography, NIOZ – Royal Netherlands Institute for Sea Research, PO Box 59, 1790 AB Den Burg, Texel, The Netherlands}

[8] {Aquatic Microbiology, Institute for Biodiversity and Ecosystem Dynamics, University of Amsterdam, P.O. Box 94248, 1090 GE, Amsterdam, The Netherlands}

Correspondence to: Alison Webb (a.l.webb@rug.nl)

**Abstract**

**The Baltic Sea is a unique environment as the largest body of brackish water in the world. Acidification of the surface oceans due to absorption of anthropogenic CO$_2$ emissions is an additional stressor facing the pelagic community of the already challenging Baltic Sea. To investigate its impact on trace gas biogeochemistry, a large-scale mesocosm experiment was performed off Tvärminne Research Station, Finland in summer 2012. During the second half of the experiment, dimethylsulphide (DMS) concentrations in the highest $f$CO$_2$ mesocosms**

**(1075 - 1333 μatm) were 34% lower than at ambient CO₂ (350 μatm). However, the net**
**production (as measured by concentration change) of seven halocarbons analysed was not**
**significantly affected by even the highest CO₂ levels after 5 weeks exposure. Methyl iodide**
**(CH₃I) and diiodomethane (CH₂I₂) showed 15% and 57% increases in mean mesocosm**
**concentration (3.8 ± 0.6 pmol L⁻¹ increasing to 4.3 ± 0.4 pmol L⁻¹ and 87.4 ± 14.9 pmol L⁻¹**
**increasing to 134.4 ± 24.1 pmol L⁻¹ respectively) during Phase II of the experiment, which**
**were unrelated to CO₂ and corresponded to 30% lower Chl-*a* concentrations compared to**
**Phase I. No other iodocarbons increased or showed a peak, with mean chloroiodomethane**
**(CH₂ClI) concentrations measured at 5.3 (± 0.9) pmol L⁻¹ and iodoethane (C₂H₅I) at 0.5 (± 0.1)**
**pmol L⁻¹. Of the concentrations of bromoform (CHBr₃; mean 88.1 ± 13.2 pmol L⁻¹),**
**dibromomethane (CH₂Br₂; mean 5.3 ± 0.8 pmol L⁻¹) and dibromochloromethane (CHBr₂Cl,**

[revised manuscript text omitted]
\text{CO}_2$ (µatm) | Mean $\text{pH}_T$ | Mean $f\text{CO}_2$ (µatm) | Mean $\text{pH}_T$ | Mean $f\text{CO}_2$ (µatm) | Mean $\text{pH}_T$ | Mean $f\text{CO}_2$ (µatm) | Mean $\text{pH}_T$ |
| M1 | Control | 331 | 7.91 | 231 | 8.00 | 328 | 7.95 | 399 | 7.86 |
| M5 | Control | 334 | 7.91 | 244 | 7.98 | 329 | 7.94 | 399 | 7.52 |
| M7 | 390 | 458 | 7.80 | 239 | 7.99 | 494 | 7.81 | 532 | 7.76 |
| M6 | 840 | 773 | 7.63 | 236 | 7.99 | 932 | 7.59 | 855 | 7.59 |
| M3 | 1120 | 950 | 7.56 | 243 | 7.98 | 1176 | 7.51 | 1027 | 7.52 |
| M8 | 1400 | 1166 | 7.49 | 232 | 8.00 | 1481 | 7.43 | 1243 | 7.45 |
| Baltic Sea | 380 | 350 | 7.91 | 298 | 7.91 | 277 | 7.98 | 436 | 7.86 |

[a] listed in order of increasing $f\text{CO}_2$

[revised manuscript text omitted]